

# Derivation of seawater pCO₂ from net community production identifies the South Atlantic Ocean as a CO₂ source

Daniel Ford[1,2], Gavin H. Tilstone[1], Jamie D. Shutler[2] and Vassilis Kitidis[1]

[1] Plymouth Marine Laboratory, Plymouth, UK

[2] College of Life and Environmental Sciences, University of Exeter, UK

*Correspondence to*: Daniel Ford (dfo@pml.ac.uk)

**Abstract.** A key step in assessing the global carbon budget is the determination of the partial pressure of $CO_2$ in seawater ($pCO_{2\,(sw)}$). Spatially complete observational fields of $pCO_{2\,(sw)}$ are routinely produced for regional and global ocean carbon budget assessments by extrapolating sparse *in situ* measurements of $pCO_{2\,(sw)}$ using satellite observations. Within these

schemes, satellite chlorophyll *a* (Chl *a*) is often used as a proxy for the biological drawdown or release of $CO_2$. Chl *a* does not however quantify carbon fixed through photosynthesis and then respired, which is determined by net community production (NCP).

In this study, $pCO_{2\,(sw)}$ over the South Atlantic Ocean is estimated using a feed forward neural network (FNN) scheme and either satellite derived NCP, net primary production (NPP) or Chl *a* to compare which biological proxy is the most accurate.

Estimates of $pCO_{2\,(sw)}$ using NCP, NPP or Chl *a* were similar, but NCP was more accurate for the Amazon Plume and upwelling regions, which were not fully reproduced when using Chl *a* or NPP. Reducing the uncertainties in the satellite biological parameters to estimate $pCO_{2\,(sw)}$, illustrated further improvement and greater differences for NCP compared to NPP or Chl *a*. Using NCP to estimate $pCO_{2\,(sw)}$ showed that the South Atlantic Ocean is a $CO_2$ source, whereas if no biological parameters are used in the FNN (following existing annual carbon assessments), this region becomes a sink for

$CO_2$. These results highlight that using NCP improved the accuracy of estimating $pCO_{2\,(sw)}$, and changes the South Atlantic Ocean from a $CO_2$ sink to a source. Reducing the uncertainties in NCP derived from satellite parameters will further improve our ability to quantify the global ocean $CO_2$ sink.

## 1.   Introduction

Since the industrial revolution, anthropogenic $CO_2$ emissions have resulted in an increase in atmospheric $CO_2$ concentrations (Friedlingstein et al., 2020; IPCC, 2013). By acting as a sink for $CO_2$, the oceans have buffered the increase in anthropogenic atmospheric $CO_2$, without which the atmospheric concentration would be 42-44 % higher (DeVries, 2014). The long-term absorption of $CO_2$ by the oceans is altering the marine carbonate chemistry of the ocean, resulting in a lowering of pH; a process known as ocean acidification (Raven et al., 2005). Observational fields of the partial pressure of $CO_2$ in seawater



($pCO_{2\,(sw)}$) are one of the key datasets needed to routinely assess the strength of the oceanic $CO_2$ sink (Friedlingstein et al., 2020; Landschützer et al., 2014, 2020; Rödenbeck et al., 2015; Watson et al., 2020b). These methods are reliant on the extrapolation of sparse *in situ* observations of $pCO_{2\,(sw)}$ using satellite observations of parameters which account for the variability of, and the controls on, $pCO_{2\,(sw)}$ (Shutler et al., 2020). These parameters include sea surface temperature (SST; e.g. Landschützer et al., 2013; Stephens et al., 1995), salinity and Chl *a* (Rödenbeck et al., 2015). SST and salinity control

$pCO_{2\,(sw)}$ by changing the solubility of $CO_2$ in seawater (Weiss, 1974), whilst biological processes such as photosynthesis and respiration contribute by modulating its concentration.

Chl *a* is routinely used as a proxy for this biological activity (Rödenbeck et al., 2015), but it does not distinguish between carbon fixation through photosynthesis and the carbon respired by the plankton community. Net primary production (the net carbon fixation rate; NPP) is determined by the standing stock of phytoplankton, for which the Chl *a* concentration is used as

a proxy, and modified by the photosynthetic rate and the available light in the water column (Behrenfeld et al., 2016). Photosynthetic rates are, in turn, modified by ambient nutrient and temperature conditions (Behrenfeld and Falkowski, 1997; Marañón et al., 2003). Elevated Chl *a* does not always equate to elevated NPP (Poulton et al., 2006), and for the same Chl *a* concentrations, NPP can vary depending on the health and metabolic state of the plankton community. All of these controls are captured by the net community production (NCP), which is the metabolic balance of the plankton community resulting

from the carbon fixed through photosynthesis and that lost through respiration. When NCP is positive the plankton community is autotrophic which implies that there is a drawdown of $CO_2$ from seawater (since the plankton reduce the $CO_2$ in the water column), and when it is negative the community is heterotrophic implying a release of $CO_2$ into the ocean (as the plankton produce or release $CO_2$) which can then be released into the atmosphere (Jiang et al., 2019; Schloss et al., 2007). Using NCP to estimate $pCO_{2\,(sw)}$ compared to Chl *a* should therefore theoretically lead to an improvement in the derivation

of $pCO_{2\,(sw)}$.

Many studies have used satellite Chl *a* to estimate $pCO_{2\,(sw)}$ at both regional (Benallal et al., 2017; Chierici et al., 2012; Moussa et al., 2016), and global scales (Landschützer et al., 2014; Liu and Xie, 2017). Chierici et al. (2012) attempted to use satellite NPP to estimate $pCO_{2\,(sw)}$ in the southern Pacific Ocean, but there was no significant improvement over using satellite Chl *a*. This is not surprising as NPP captures more of the biological signal, but still lacks any inclusion of respiration which results in the release of $CO_2$ into the water column. To our knowledge the use of satellite NCP to estimate $pCO_{2\,(sw)}$

has not been attempted before and could be a means of improving estimates of $pCO_{2\,(sw)}$ as long as satellite NCP observations are accurate (Ford et al., 2021; Tilstone et al., 2015a). These satellite measurements may improve the estimation of $pCO_{2\,(sw)}$ as NCP includes the full biological control on $pCO_{2\,(sw)}$, which could be important in regions of sparse *in situ* $pCO_{2\,(sw)}$ observations where interpolation and neural network techniques are likely to struggle (Watson et al., 2020b).

The objective of this paper is therefore to compare the estimation of $pCO_{2\,(sw)}$ using either NCP, NPP or Chl *a* to determine which biological descriptor is the most accurate. A 16 year time series of $pCO_{2\,(sw)}$ was generated for the South Atlantic Ocean using satellite NCP, NPP or Chl *a*, as the biological input, alongside a baseline approach with no biological parameters as input. Regional differences in the generated $pCO_{2\,(sw)}$ fields are assessed. The seasonal and interannual





variability in pCO$_{2\,(sw)}$ estimated from NCP, NPP, Chl *a* and the baseline approach were also compared. A perturbation
analysis was conducted to evaluate the potential reduction in the uncertainty in the pCO$_{2\,(sw)}$ fields when estimated from
NCP, NPP or Chl *a*. This is discussed in the context of reducing uncertainties in these input variables for future
improvements in producing spatially complete fields of pCO$_{2\,(sw)}$, and the effect on estimates of the oceanic carbon sink.

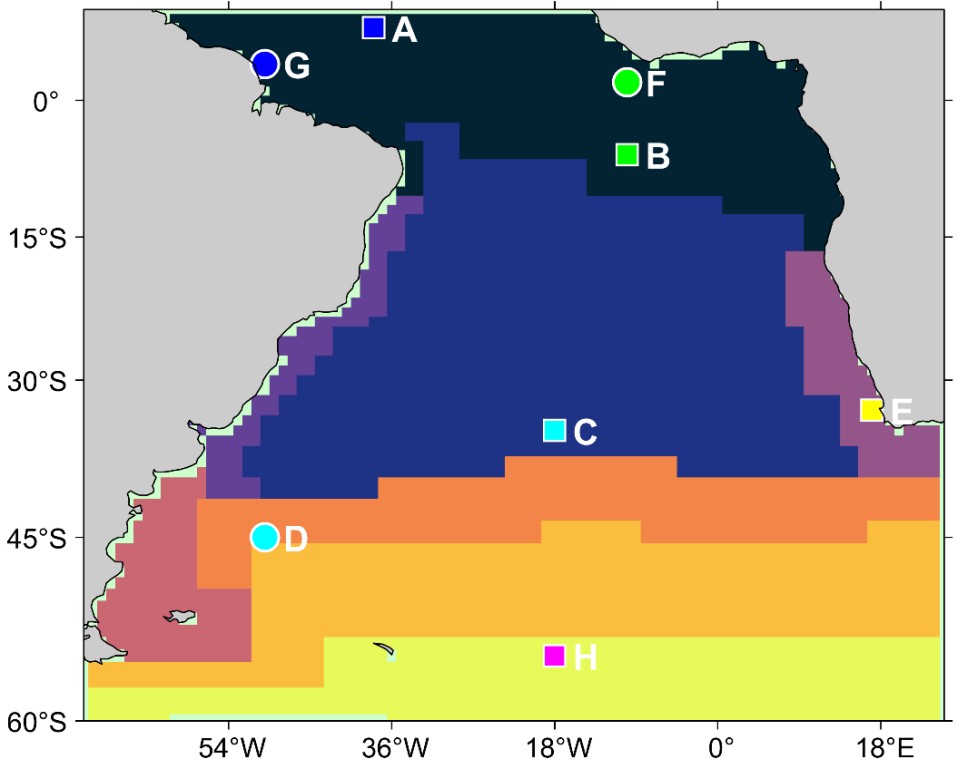

**Fig. 1: Map of the 8 static biogeochemical provinces in the South Atlantic Ocean. Markers and letters indicate the locations of**
**timeseries extracted in Fig. 3.**



## 2. Methods

### 2.1. Surface Ocean Carbon Atlas (SOCAT) pCO$_2$ $_{(sw)}$ and atmospheric CO$_2$

SOCATv2020 (Bakker et al., 2016; Pfeil et al., 2013) individual fugacity of CO$_2$ in seawater (fCO$_2$ $_{(sw)}$) observations were
downloaded from https://www.socat.info/index.php/data-access/. Data were extracted from 2002 to 2018 for the South
Atlantic Ocean (10º N-60º S, 25º E–80º W; Fig. 1). The individual cruise observations were collected from different depths,
and are not representative of the fCO$_2$ $_{(sw)}$ in the top ~100 μm of the ocean, where gas exchange occurs (Goddijn-Murphy et
al., 2015; Woolf et al., 2016). Therefore, the SOCAT observations were re-analysed to a standard temperature dataset and
depth (Reynolds et al., 2002) that is considered representative of the bottom of the mass boundary layer (Woolf et al., 2016).
This was achieved using the 'fe_reanalyse_socat' utility in the open source FluxEngine toolbox (Holding et al., 2019;
Shutler et al., 2016), which follows the methodology described in Goddijn-Murphy et al. (2015). The reanalysed fCO$_2$ $_{(sw)}$
observations were converted to pCO$_2$ $_{(sw)}$, and gridded onto 1º monthly grids following SOCAT protocols (Sabine et al.,
2013). The uncertainties in the *in situ* data were taken as the standard deviation of the observations in each grid cell, or
where a single observation exists were set as 5 μatm following Bakker et al. (2016).
Monthly 1º grids of atmospheric pCO$_2$ (pCO$_2$ $_{(atm)}$) were extracted from v5.5 of the global estimates of pCO$_2$ $_{(sw)}$ dataset
(Landschützer et al., 2016, 2017). pCO$_2$ $_{(atm)}$ was estimated using the dry mixing ratio of CO$_2$ from the NOAA-ESRL marine
boundary layer reference (https://www.esrl.noaa.gov/gmd/ccgg/mbl/), Optimum Interpolated SST (Reynolds et al., 2002)
and sea level pressure following Dickson et al. (2007).

### 2.2. Moderate Resolution Spectroradiometer on Aqua (MODIS-A) satellite observations

4 km resolution monthly mean Chl *a* were calculated from MODIS-A Level 1 granules, retrieved from National Aeronautics
and Space Administration (NASA) Ocean Colour website (https://oceancolor.gsfc.nasa.gov/) using SeaDAS v7.5, and
applying the standard OC3-CI Chl *a* algorithm (https://oceancolor.gsfc.nasa.gov/atbd/chlor_a/). In addition, monthly mean
MODIS-A SST and photosynthetically active radiation (PAR) were also downloaded from the NASA Ocean Colour website.
Mean monthly NPP were generated from MODIS-A Chl *a*, SST and PAR using the Wavelength Resolving Model (Morel,
1991) with the look up table described in Smyth et al. (2005). Coincident mean monthly NCP using the algorithm NCP-D
described in Tilstone et al. (2015a) were generated using the MODIS-A NPP and SST data. Further details of the satellite
algorithms are given in O'Reilly et al. (1998; 2019), Hu et al. (2012) for Chl *a*, Smyth et al. (2005), Tilstone et al. (2005,
2009) for NPP and Tilstone et al. (2015a) for NCP. All monthly mean data were generated between July 2002 and December
2018 and were re-gridded onto the same 1º grid as the pCO$_2$ $_{(sw)}$ observations. The assessed uncertainties from the literature
for each of the input parameters used are given in Table 1.





**Table 1: Uncertainties in the input parameters of the Feed Forward Neural Network used in Monte Carlo uncertainty propagation and perturbation analysis.**

| Parameter | Algorithm Uncertainty | Reference |
|---|---|---|
| Chlorophyll $a$ | 0.15 $\log_{10}$(mg m$^{-3}$) | Ford, et al (2021) |
| Net Primary Production | 0.20 $\log_{10}$(mg C m$^{-2}$ d$^{-1}$) | Ford, et al (2021) |
| Net Community Production | 45 mmol O$_2$ m$^{-2}$ d$^{-1}$ | Ford, et al (2021) |
| SST | 0.41 ℃ | Ford, et al (2021) |
| pCO$_{2\ (atm)}$ | 1 µatm | |


## 2.3. Feed forward neural network scheme

The South Atlantic Ocean was partitioned into 8 biogeochemical provinces (Fig. 1), following Longhurst et al. (1995). The pCO$_{2\ (sw)}$ observations in the eastern Equatorial Atlantic were sparse, and therefore the Equatorial region was merged into 1 province. In each province the available monthly pCO$_{2\ (sw)}$ observations were matched to temporally and spatially coincident

pCO$_{2\ (atm)}$, MODIS-A, NCP and SST, to provide training data for the feed forward neural network (FNN). Observations in coastal regions (< 200 m water depth) were removed from the analysis, due to the increased uncertainty in ocean colour observations in these areas (e.g. Lavender et al., 2004). Due to constraints on the coverage of ocean colour data, no data were available in austral winter below ~50º S.

The coincident observations in each province were randomly split into 3 datasets: 1.) A training dataset (50 % of the

observations) used to train the FNNs; 2.) A validation dataset (30 % of the observations) used to assess the performance of the FNN and to prevent the networks from overfitting; 3.) An independent test dataset (20 % of the observations) to assess the final performance of the FNN, with observations that are independent of the network training. The optimal split (r$_{opt}$) method of Amari et al. (1997) was used to partition the data into these three sets, as follows:

$$r_{opt} = 1 - \frac{1}{\sqrt{2m}} \qquad (1)$$

where m is number of input parameters. For our three input parameters, a split of 60 % training to 40 % validation datasets would occur, where we removed 10 % from each dataset to provide a further independent test dataset. A pre-training step was used to determine the optimum number of hidden neurons in the FNN (Benallal et al., 2017; Landschützer et al., 2013; Moussa et al., 2016), to provide the best fit for the observations, whilst preventing over fitting (Demuth et al., 2008).

The FNN was trained using the optimum number of hidden neurons, in an iterative process until the Root Mean Square

Difference (RMSD) remained unchanged for 6 iterations. The best performing FNN, with the lowest RMSD was then used to estimate pCO$_{2\ (sw)}$. The uncertainties in the input parameters were propagated through the FNN, using a Monte Carlo uncertainty propagation, where 1000 calculations were made perturbing the input parameters, using random noise for their uncertainty (Table 1). The output from the 8 province FNNs were then combined and weighted statistics, which account for





both the satellite and *in situ* uncertainty, were used to assess the overall performance of the FNN (as used within Ford et al.,

2021). The combined 8 FNNs approach will hereafter be referred to as SA-FNN.

The approach to training the FNNs was repeated replacing NCP with Chl *a* or NPP sequentially, to determine if there was an improvement by using NCP. A baseline SA-FNN with no biological parameters as input was trained using $pCO_{2\,(atm)}$, MODIS-A SST, alongside sea surface salinity and mixed layer depth from the Copernicus Marine Environment Modelling Service (https://resources.marine.copernicus.eu/) global ocean physics reanalysis product (GLORYS12V1). This parameter

combination has recently been included within a neural network scheme to estimate global fields of $pCO_{2\,(sw)}$ (Watson et al., 2020b).

Following these methods, a monthly mean time-series of $pCO_{2\,(sw)}$ was generated in the South Atlantic Ocean, applying the SA-FNN approach using NCP (SA-FNN_NCP), NPP (SA-FNN_NPP), Chl *a* (SA-FNN_CHLA) or no biological parameters (SA-FNN_NO-BIO). The $pCO_{2\,(sw)}$ fields were spatially averaged using a 3×3 pixel filter, but were not averaged temporally as in

previous studies (Landschützer et al., 2014, 2016) because averaging temporally could mask features that occur within single months of the year. The uncertainties in the input parameters (Table 1) were propagated through the neural network on a per pixel basis, and combined in quadrature with the RMSD of the test dataset, to produce a combined uncertainty budget for each pixel, assuming all sources of uncertainty are independent and uncorrelated (BIPM, 2008; Taylor, 1997).

### 2.4. Atlantic Meridional Transect *in situ* data

To assess the accuracy of the SA-FNN, coincident *in situ* measurements of NCP, NPP, Chl *a*, SST, $pCO_{2\,(atm)}$ and $pCO_{2\,(sw)}$, with uncertainties, were provided by Atlantic Meridional Transects 20, 21, 22 and 23 in 2010, 2011, 2012 and 2013, respectively. All the Atlantic Meridional Transect data described in this section can be obtained from the British Oceanographic Data Centre (https://www.bodc.ac.uk/). Chl *a* was estimated following the methods of Brewin et al. (2016), using underway continuous spectrophotometric measurements, and uncertainties were estimated as ~0.06 $\log_{10}(mg\ m^{-3})$

(Ford et al., 2021). $^{14}C$ based NPP measurements were made based on dawn to dusk simulated *in situ* incubations, following the methods given in Tilstone et al. (2017), at 56 stations with a per station uncertainty. Uncertainties ranged between 8 and 213 mg C m$^{-2}$ d$^{-1}$ and were on average 53 mg C m$^{-2}$ d$^{-1}$. NCP was estimated using *in vitro* changes in dissolved $O_2$, following the methods of Gist et al. (2009) and Tilstone et al. (2015a) at 51 stations with a per station uncertainty calculated. Uncertainties ranged between 5 and 25 mmol $O_2$ m$^{-2}$ d$^{-1}$ and were on average 14 mmol $O_2$ m$^{-2}$ d$^{-1}$.

Underway measurements of $pCO_{2\,(sw)}$ and $pCO_{2\,(atm)}$ were performed continuously, following the methods of Kitidis et al. (2017). SST was continuously measured alongside all observations (SeaBird SBE45), with a factory calibrated uncertainty of ±0.01 °C. The mean of underway $pCO_{2\,(sw)}$, $pCO_{2\,(atm)}$, SST and Chl *a* were taken ±20 minutes around each station where NCP and NPP were measured.





### 2.5. Perturbation analysis

Following the approach of Saba et al. (2011), a perturbation analysis was conducted, to evaluate the potential reduction in SA-FNN $pCO_{2 \text{ (sw)}}$ RMSD that could be attributed to the input parameters. The analysis indicates the maximum reduction in RMSD that could be achieved if uncertainties in the input parameters were reduced to ~0. Each of the input parameters; NCP, SST and $pCO_{2 \text{ (atm)}}$ can have three possible values for each *in situ* $pCO_{2 \text{ (sw)}}$ estimate (original value, original ± uncertainty; Table 1), enabling 27 perturbations of the input data. RMSD and bias were used to assess the performance of

SA-FNN under different scenarios. For each *in situ* $pCO_{2 \text{ (sw)}}$ observation, the 27 perturbations of SA-FNN $pCO_{2 \text{ (sw)}}$ were examined, and the perturbation that produced the lowest RMSD and bias combination in two scenarios was selected; (1) uncertainty in individual input parameters (NCP, SST and $pCO_{2 \text{ (atm)}}$) and (2) uncertainty in all input parameters together. The approach was conducted on all three training datasets, and on the Atlantic Meridional Transect *in situ* data. The analysis was repeated sequentially replacing NCP with Chl *a* and NPP, to determine if there was a greater maximum reduction in

RMSD using NCP. The analysis was also conducted allowing for a 10 % reduction in input parameter uncertainties, to indicate the short-term reduction in $pCO_{2 \text{ (sw)}}$ RMSD that could be achieved by reducing the input parameter uncertainties.

### 2.6. Comparison of the SA-FNN_{NCP} with the SA-FNN_{NO-BIO}, SA-FNN_{CHLA}, SA-FNN_{NPP} and 'state of the art' data for the South Atlantic

The most comprehensive $pCO_{2 \text{ (sw)}}$ fields to date are from Watson et al. (2020b, 2020a). The 'standard method' $pCO_{2 \text{ (sw)}}$

fields within the Watson et al. (2020b, 2020a) data were produced by extrapolating the *in situ* reanalysed SOCATv2019 $pCO_{2 \text{ (sw)}}$ observations using a self-organising map feed forward neural network approach (Landschützer et al., 2016), and will be referred to as 'W2020'. A time-series was extracted from the W2020 data, coincident to SA-FNN_{NCP}, SA-FNN_{NPP}, SA-FNN_{CHLA} and SA-FNN_{NO-BIO}. For the five methods, a monthly climatology referenced to the year 2010 was computed, assuming an atmospheric $CO_2$ increase of 1.5 µatm yr$^{-1}$ (Takahashi et al., 2009; Zeng et al., 2014). The climatology should

be insensitive to the chosen atmospheric $CO_2$ rise due to the reference year being central to the time series. The standard deviation of this climatology was also computed on a per pixel basis.

For each station in Fig. 1, the monthly climatology of $pCO_{2 \text{ (sw)}}$, representing the average seasonal cycle of $pCO_{2 \text{ (sw)}}$, and the standard deviation of the climatology, as an indication of the interannual variability, were extracted from the five approaches. The $pCO_{2 \text{ (sw)}}$ value for each station was the statistical mean of the four nearest data points weighted by their

respective proximity to the station coordinate.

The station climatologies for the SA-FNN_{NO-BIO}, W2020, SA-FNN_{CHLA}, and SA-FNN_{NPP} were compared to the SA-FNN_{NCP}, by testing for significant differences in the seasonal cycle and annual $pCO_{2 \text{ (sw)}}$ (offset). The seasonal cycles were compared using a non-parametric Spearman's correlation and deemed statistically different where the correlation was not significant ($\alpha < 0.05$). A non-parametric Kruskal-Wallis tested for significant ($\alpha < 0.05$) differences in the annual $pCO_{2 \text{ (sw)}}$, indicating an

offset between the two tested climatologies. The Southern Ocean station (station H) was excluded from the statistical analysis due to missing data in the SA-FNN.



### 2.7. Estimation of the bulk CO$_2$ flux

The flux of CO$_2$ (F) between the atmosphere and ocean (air-sea) can be expressed in a bulk parameterisation as:

$$F = k \left( \alpha_W \, pCO_{2\,(sw)} - \alpha_s \, pCO_{2\,(atm)} \right) \tag{2}$$

Where k is the gas transfer velocity, and $\alpha_w$ and $\alpha_s$ are the solubility of CO$_2$ at the base and top of the mass boundary layer at the sea surface respectively (Woolf et al., 2016). k was estimated from ERA5 monthly reanalysis wind speed (downloaded from the Copernicus Climate Data Store; https://cds.climate.copernicus.eu/) following the parameterisation of Nightingale et al. (2000). $\alpha_w$ was estimated as a function of SST and sea surface salinity (Weiss, 1974) using the monthly Optimum Interpolated SST (Reynolds et al., 2002) and sea surface salinity from the Copernicus Marine Environment Modelling

Service global ocean physics reanalysis product (GLORYS12V1). $\alpha_s$ was estimated using the same temperature and salinity datasets but included a gradient from the base to the top of mass boundary layer of -0.17 K (Donlon et al., 1999) and +0.1 salinity units (Woolf et al., 2016). pCO$_2$ $_{(atm)}$ was estimated using the dry mixing ratio of CO$_2$ from the NOAA-ESRL marine boundary layer reference, Optimum Interpolated SST (Reynolds et al., 2002) applying a cool skin bias (0.17K; Donlon et al., 1999) and sea level pressure following Dickson et al. (2007). Spatially and temporally complete pCO$_2$ $_{(sw)}$ fields, which are

representative of pCO$_2$ $_{(sw)}$ at the base of the mass boundary layer, were extracted from the SA-FNN$_{NCP}$, SA-FNN$_{NPP}$, SA-FNN$_{CHLA}$, SA-FNN$_{NO-BIO}$ and W2020.

The monthly CO$_2$ flux was calculated using the open source FluxEngine toolbox (Holding et al., 2019; Shutler et al., 2016) between 2003 and 2018 for the five pCO$_2$ $_{(sw)}$ inputs, using the 'rapid transport' approximation (described in Woolf et al., 2016). The net annual flux was determined for the South Atlantic Ocean (10° N-44° S; 25° E-70° W) using the

'fe_calc_budgets.py' utility within FluxEngine with the supplied area and land percentage masks. The mean net annual flux was calculated as the mean of the 15 year net annual fluxes. Positive net fluxes indicate a net source to the atmosphere, and negative net fluxes a sink.

**Table 2: The percentage reduction in pCO$_2$ $_{(sw)}$ RMSD by reducing NCP, net primary production and chlorophyll a uncertainties**
**to ~0 as described in Section 2.5. The full results can be found in Appendix Table A1.**

| Parameter | Training | Validation | Independent Test | AMT *in situ* |
|---|---|---|---|---|
| NCP | 32 % | 40 % | 36 % | 25 % |
| Net Primary Production | 31 % | 37 % | 36 % | 13 % |
| Chlorophyll *a* | 17 % | 21 % | 20 % | 7 % |





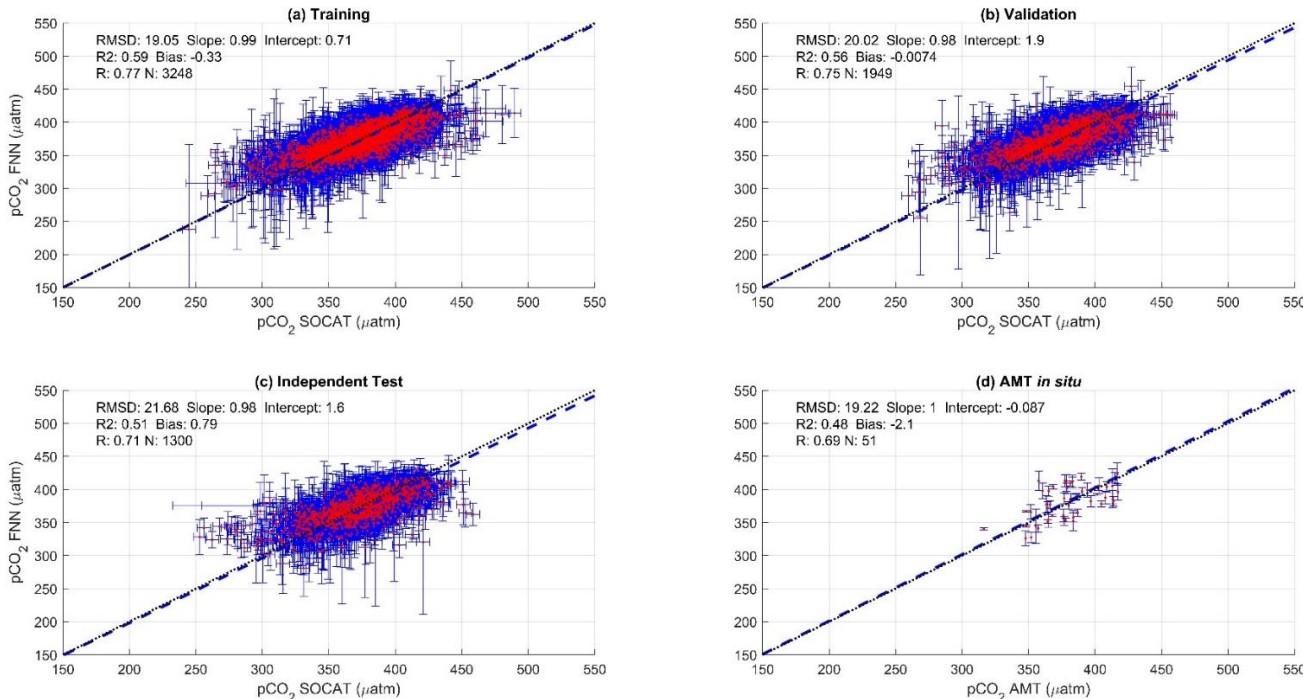

**Fig. 2: Scatter plots showing the combined performance of the 8 feed forward neural networks trained using NCP for each**
**biogeochemical province (Fig. 1) using 4 separate training and validation datasets; (a) Training, (b) Validation, (c) Independent Test and (d) Atlantic Meridional Transect (AMT) in situ. The blue dashed line is the Type II regression and the black dashed line is the 1:1 line. Horizontal errorbars indicate the uncertainty of the SOCATv2020 pCO$_2$ (sw). Vertical errorbars indicate the uncertainty attributed to the input parameter uncertainty propagated through the feed forward neural networks. The statistics within each plot are; Root Mean Square Difference (RMSD), Slope and Intercept of the Type II regression, Coefficient of**
**Determination (R2), Pearson's Correlation Coefficient (R), Bias and number of samples (N).**

*Table 3: The percentage reduction in pCO$_2$ (sw) RMSD by reducing NCP, net primary production and chlorophyll a uncertainties by 10 % as described in Section 2.5.*

| Parameter | Training | Validation | Independent Test | AMT *in situ* |
|---|---|---|---|---|
| NCP | 7 % | 8 % | 8 % | 3 % |
| Net Primary Production | 5 % | 6 % | 5 % | 1.5 % |
| Chlorophyll *a* | 2 % | 2 % | 2 % | 0.5 % |





## 3.  Results

### 3.1. SA-FNN performance and perturbation analysis

The performance of the SA-FNN trained using $pCO_{2 (atm)}$, SST and NCP for the three training datasets is given in Fig. 2. The SA-FNN$_{NCP}$ had an accuracy (RMSD) of 21.67 µatm and a precision (bias) of 0.87 µatm, which was determined with the independent test data (N = 1300). Training the SA-FNN using Chl *a* or NPP instead of NCP, resulted in a similar performance (Appendix A Fig. A1, Fig. A2). The RMSD for the independent test data was within ~1.5 µatm for Chl *a* (19.88

µatm), NPP (20.48 µatm) and NCP (21.68 µatm) and bias near zero.

The reduction in $pCO_{2 (sw)}$ RMSD that could be achieved if input parameter uncertainties were reduced to ~0 was assessed using the previously described perturbation analysis (Table 2, Appendix A Table A1). This showed that satellite NCP uncertainties lead to an on average 36 % reduction in $pCO_{2 (sw)}$ RMSD, with NPP a 34 % reduction and Chl *a* a 19 % reduction, indicating that improving NCP uncertainties will have the largest impact on improving the estimated $pCO_{2 (sw)}$

fields. The bias remained near zero for all parameters indicating good precision of the SA-FNN approach (not shown). Applying the Atlantic Meridional Transect *in situ* data as input to the SA-FNN and resulting perturbation analysis, a decrease in $pCO_{2 (sw)}$ RMSD of 25 % for NCP, 13 % for NPP and 7 % for Chl *a* was observed.

The reduction in $pCO_{2 (sw)}$ RMSD from reducing input parameter uncertainties by 10 % was also assessed through the perturbation analysis (Table 3). This indicated a decrease in $pCO_{2 (sw)}$ RMSD of 8 % for NCP, 5 % for NPP and 2 % for Chl

*a*, again indicating that improving NCP uncertainties will have the largest impact on improving the estimated $pCO_{2 (sw)}$ fields.





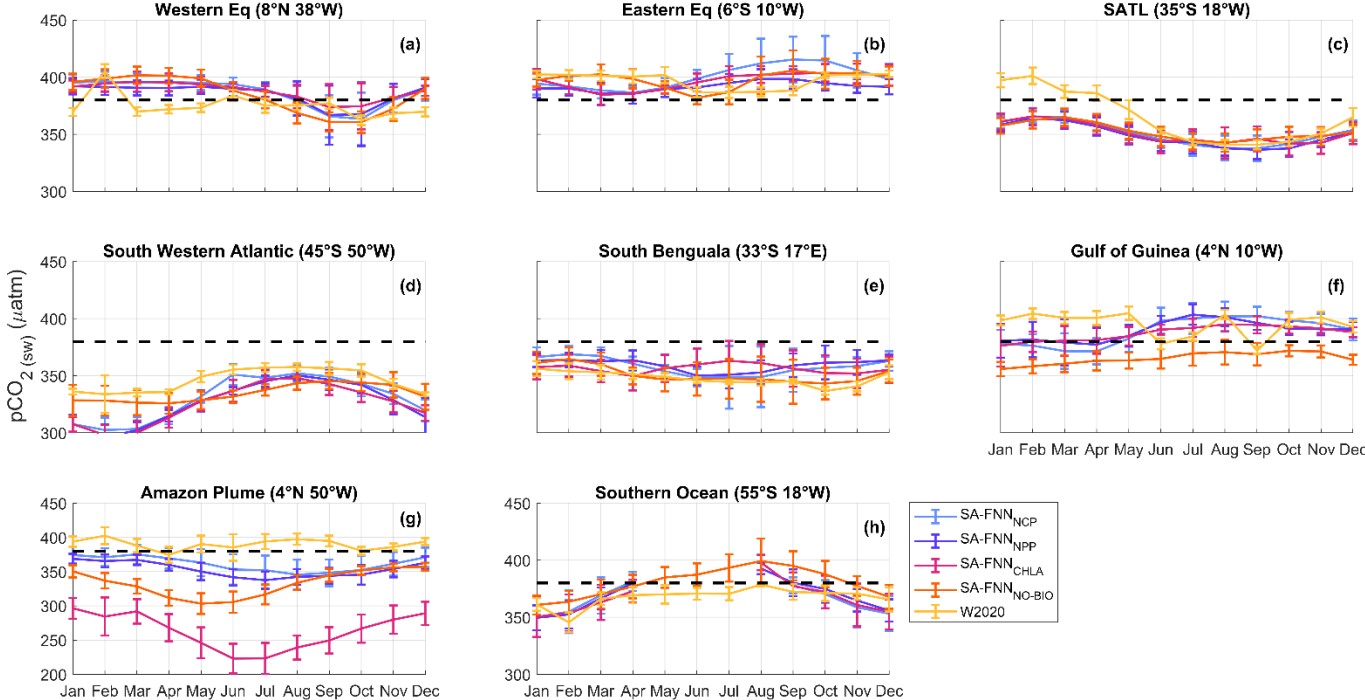

**Fig. 3: Monthly climatologies of pCO$_{2\,(sw)}$ referenced to the year 2010 for the 8 stations marked in Fig. 1 from the SA-FNN$_{NCP}$, SA-FNN$_{NPP}$, SA-FNN$_{CHLA}$, SA-FNN$_{NO\text{-}BIO}$ and W2020 (Watson et al., 2020b). The atmospheric CO$_2$ increase was set as 1.5 µatm yr$^{-1}$. Black dashed line indicates the atmospheric pCO$_2$ (~380 µatm). Errorbars indicate the standard deviation of the climatology, where larger errorbars indicate a larger interannual variability.  Note the different y-axis limits in Fig. 3g.**

## 3.2. Comparison of SA-FNN$_{NCP}$ with other methods

The monthly climatology of pCO$_{2\,(sw)}$ generated using the SA-FNN$_{NCP}$ and referenced to the year 2010 showed differences with two published climatologies, especially in the Equatorial region (Appendix B). The monthly climatology for 8 stations (Fig. 1) were extracted from the SA-FNN$_{NCP}$, SA-FNN$_{NPP}$, SA-FNN$_{CHLA}$, SA-FNN$_{NO\text{-}BIO}$ and the W2020, to assess differences between the pCO$_{2\,(sw)}$ estimates (Fig. 3). The SA-FNN$_{NCP}$ and SA-FNN$_{NO\text{-}BIO}$ showed significant divergences in the Equatorial Atlantic (Figs. 3b, f, g; Fig. 4). At the eastern equatorial station, the interannual variability in pCO$_{2\,(sw)}$ from the SA-FNN$_{NCP}$ was high and a minimum occurred between January and April, which slowly increased to a maximum in September and October (Fig. 3b). The SA-FNN$_{NO\text{-}BIO}$ showed the opposite pattern, with a pCO$_{2\,(sw)}$ minima between May to July and a maxima for the remaining months with little interannual variability. The Gulf of Guinea station showed a similar variability in the SA-FNN$_{NCP}$ pCO$_{2\,(sw)}$ except the maxima was lower at this station (Fig. 3f). The SA-FNN$_{NO\text{-}BIO}$ indicated pCO$_{2\,(sw)}$ below the SA-FNN$_{NCP}$ throughout the year. The greatest divergence occurred near the Amazon plume (Fig. 3g)





where SA-FNN$_{NCP}$ pCO$_{2\,(sw)}$ was below or at pCO$_{2\,(atm)}$ for all months and there was a large interannual variability in pCO$_2$ $_{(sw)}$. The SA-FNN$_{NO-BIO}$ displayed lower pCO$_{2\,(sw)}$ between January and July and a lower interannual variability (Fig. 3g).

The SA-FNN$_{NCP}$ and SA-FNN$_{NO-BIO}$ showed no significant difference in the seasonal patterns of pCO$_{2\,(sw)}$ at stations south of 20 °S (Figs. 3c, d, e; Fig. 4). There was, however, a significant offset at some stations where the SA-FNN$_{NCP}$ generally exhibited lower pCO$_{2\,(sw)}$ in austral summer and a higher interannual variation. The SA-FNN$_{NCP}$ was significantly different to the W2020 at similar stations as the SA-FNN$_{NO-BIO}$ (Fig. 3, Fig. 4).

The SA-FNN$_{NCP}$ and SA-FNN$_{CHLA}$ showed significant differences in pCO$_{2\,(sw)}$ values in the South Benguela and Amazon
Plume. In the South Benguela (Fig. 3e; Fig. 4), SA-FNN$_{NCP}$ gave a pCO$_{2\,(sw)}$ maxima in austral summer, whereas the SA-FNN$_{CHL}$ maximum occurred in austral winter. In the Amazon Plume there was significant offset between the two methods and the SA-FNN$_{CHL}$ gave lower pCO$_{2\,(sw)}$ compared to the SA-FNN$_{NCP}$ (Fig. 3g; Fig. 4). The SA-FNN$_{NCP}$ and SA-FNN$_{NPP}$ had a significant offset at the Eastern Equatorial station (Fig. 3c; Fig. 4), where the SA-FNN$_{NPP}$ indicated lower pCO$_{2\,(sw)}$. For the other stations, no significant differences were observed.


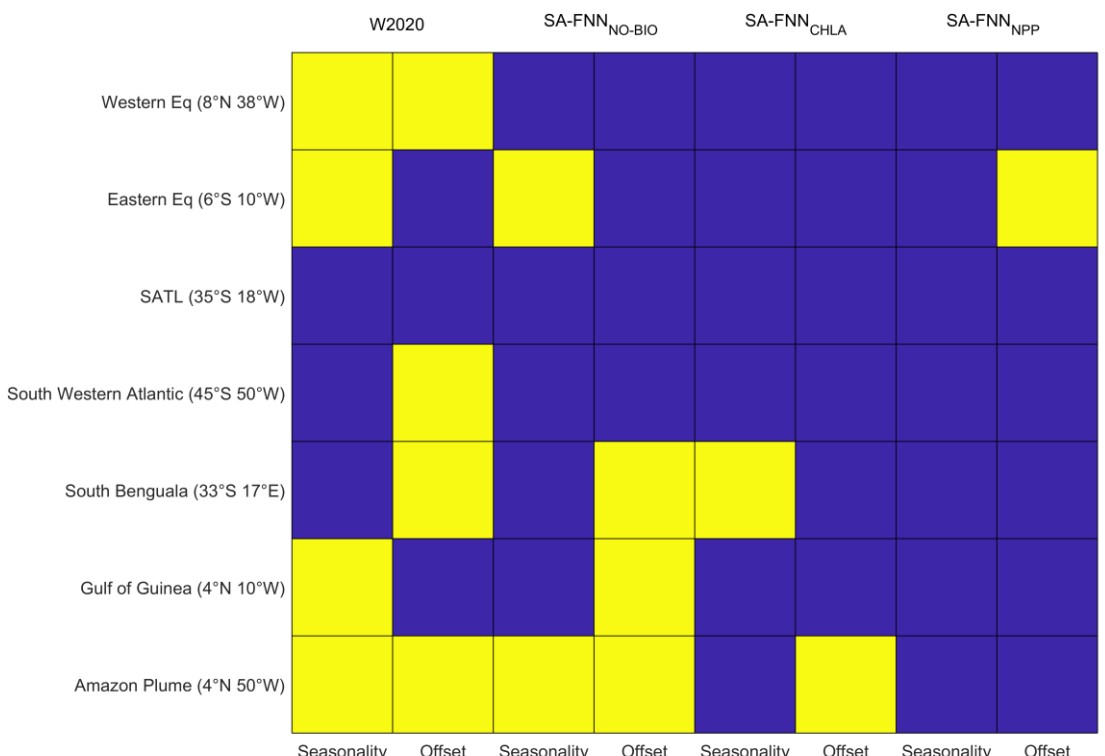

**Fig. 4: Statistical comparison of the SA-FNN$_{NCP}$ with the W2020, SA-FNN$_{NO-BIO}$ SA-FNN$_{CHLA}$ and SA-FNN$_{NPP}$ climatologies, where yellow blocks indicate a significant difference (α = 0.05). Seasonality indicates a difference in the seasonal cycle and offset indicates a difference between the mean pCO$_{2\,(sw)}$ of the climatologies.**



## 4. Discussion

### 4.1. Assessment of biological parameters to estimate pCO$_2$ (sw)

In this paper, the differences in estimating pCO$_2$ (sw) using satellite NCP, NPP or Chl $a$ were assessed. The SA-FNN$_{NCP}$ had an overall accuracy that is consistent with other approaches that have been developed for the Atlantic (22.83 µatm; Landschützer et al., 2013), and slightly lower than the published global result of 25.95 µatm (Landschützer et al., 2014). Training the SA-FNN using Chl $a$, NPP or NCP, there was no overall improvement in the broad-scale accuracy of pCO$_2$ (sw) compared to literature values in the South Atlantic. When the uncertainties in the input parameters were investigated however, differences in the estimates of pCO$_2$ (sw) were apparent. The perturbation analysis indicated that up to a 36 % improvement in estimating pCO$_2$ (sw) could be achieved if NCP data uncertainties were reduced. A similar improvement could be obtained if the NPP uncertainties were reduced. Ford et al. (2021) showed that up to 40 % of the uncertainty in satellite NCP is attributed to the uncertainty in satellite NPP, which is an input to the NCP approach. This suggests that improvements in estimating NPP from satellite data will lead to a further improvement in estimating pCO$_2$ (sw) from NCP. These improvements could be achieved through advances in the water column light field (e.g. Sathyendranath et al., 2020) or assignment of photosynthetic parameters (e.g. Kulk et al., 2020), for example. For a discussion on improving satellite NPP estimates we refer the reader to Lee et al. (2015).

Satellite NCP is reliant on NPP as input (Tilstone et al., 2015a; Ford et al., 2021), and NPP requires Chl $a$ as the primary input (e.g. Lobanova et al., 2018; Platt et al., 1991; Tilstone et al., 2015b). To uncouple the Chl $a$, NPP and NCP estimates and their uncertainties, the perturbation analysis was also conducted on Atlantic Meridional Transect $in$ $situ$ observations. This showed that reducing $in$ $situ$ NCP uncertainties provided the greatest reduction in pCO$_2$ (sw) RMSD, which was three times the reduction achievable using Chl $a$. This indicates that the optimal predictive power of Chl $a$ to estimate pCO$_2$ (sw) has been reached and to achieve further improvements in estimates of pCO$_2$ (sw) and reduction in its associated uncertainty, requires the use of NCP.

A reduction of input uncertainties to ~0 is near impossible, but a reduction by 10 % could be feasible (e.g. NCP uncertainty reduced from 45 to 40.5 mmol O$_2$ m$^{-2}$ d$^{-1}$; Table 1). A perturbation analysis conducted for this showed similar results, with NCP producing the greatest reduction in pCO$_2$ (sw) RMSD of 8 % compared to 2 % for Chl $a$. Thus reducing NCP uncertainties will provide a greater improvement in pCO$_2$ (sw) compared to reducing the uncertainties in Chl $a$.

These improvements in estimating NCP could be achieved through many components. Ford et al. (2021) showed 40 % of satellite NCP uncertainties were attributed to $in$ $situ$ NCP uncertainties. The $in$ $situ$ bottle incubation measurements could be improved using the principles of Fiducial Reference Measurements (FRM; Banks et al., 2020), which are traceable to metrology standards, referenced to inter-comparison exercises, with a full uncertainty budget. This however, becomes complicated when considering the number of different methods to measure NCP and the large divergence between them (Robinson et al., 2009). A review of these methods has already been conducted (Duarte et al., 2013; Ducklow and Doney, 2013; Williams et al., 2013). The methods broadly fall into the following categories: a.) $in$ $vitro$ incubations of samples





under light/dark treatments (Gist et al., 2009) and b.) *in situ* observations of oxygen to argon ($O_2/Ar$) ratios (Kaiser et al., 2005) or the observed isotopic signature of oxygen (Kroopnick, 1980; Luz and Barkan, 2000). All of these methods are

subject to, but do not account for, the photochemical sink which may lead to underestimation of *in vitro* NCP by up to 22 % (Kitidis et al., 2014). Independent ground measurements that use accepted protocols for the *in vitro* method are currently made on the Atlantic Meridional Transect, however a community consensus should consider a consistent methodology for NCP. Increasing the number of such observations for the purpose of algorithm development, would further constrain the NCP, but also provide observations across the lifetime of newly launched satellites. The uncertainties on each *in vitro*

measurement are assessed through replicate bottles which could be used to calculate a full uncertainty budget for each NCP measurement when combined with analytical uncertainties.

Serret et al. (2015) indicated that NCP is controlled by both the heterogeneity in NPP and respiration. The satellite NCP algorithm applied in this study accounts for some of the heterogeneity in respiration, through an empirical SST to NCP relationship (Tilstone et al., 2015a). Quantifying the variability in respiration could further improve NCP estimates when

coupled with NPP rates from satellite observations.

### 4.2. Accuracy of SA-FNN$_{NCP}$ pCO$_{2 (sw)}$ at seasonal and interannual scales

The seasonal and interannual variability of pCO$_{2 (sw)}$ estimated using the SA-FNN$_{NCP}$ was compared with the SA-FNN$_{NO-BIO}$, W2020 (Watson et al., 2020b), SA-FNN$_{CHL}$ and SA-FNN$_{NPP}$ at 8 stations. The stations (Fig. 1) represent locations of previous studies into *in situ* pCO$_{2 (sw)}$ variability in the South Atlantic Ocean and allow comparisons with literature values.

Significant differences between the SA-FNN$_{NCP}$ and SA-FNN$_{NO-BIO}$ were observed at four stations (Fig. 4), especially in the Equatorial Atlantic.

At 8° N 38° W (Fig. 3a), Lefèvre et al. (2020) reported pCO$_{2 (sw)}$ to be stable at ~400 µatm, between June and August 2013, and to decrease in September to ~360 µatm, which is attributed to the Amazon Plume propagating into the western Equatorial Atlantic (Coles et al., 2013). Bruto et al. (2017) indicated however, that elevated pCO$_{2 (sw)}$ at ~430 µatm exist

throughout the year from 2008 to 2011. For the station in the Amazon Plume at 4° N 50° W (Fig. 3g), where the effects of the plume extend northwest towards the Caribbean (Coles et al., 2013; Varona et al., 2019), Lefèvre et al. (2017) indicated that this region acts as a sink for CO$_2$ (pCO$_{2 (sw)}$ < pCO$_{2 (atm)}$), especially between May to July, coincident with maximum discharge from the Amazon River (Dai and Trenberth, 2002). Valerio et al. (2021) indicated pCO$_{2 (sw)}$ varied above and below pCO$_{2 (atm)}$ at 4° N 50° W consistent with the SA-FNN$_{NCP}$. The interannual variability of pCO$_{2 (sw)}$ has been shown to be

high in this region in all months (Lefèvre et al., 2017). The SA-FNN$_{NCP}$ provided a better representation of the seasonal and interannual variability induced by the Amazon River discharge and associated plume at these two stations compared to the SA-FNN$_{NO-BIO}$, although differences were small at 8° N 38° W.

The station in the Eastern Tropical Atlantic at 6° S 10° W (Fig. 3b), is under the influence of the equatorial upwelling (Lefèvre, Guillot, Beaumont, & Danguy, 2008), which is associated with upwelling of CO$_2$ rich waters between June and

September. Lefèvre et al. (2008) indicated that peak pCO$_{2 (sw)}$ of ~440 µatm was observed in September, and remained stable





until December, before decreasing to a minima of ~360 µatm in May (Parard et al., 2010). Lefèvre et al. (2016) showed however, that the influence of the equatorial upwelling does not reach the buoy in all years, and in some years lower $pCO_{2\ (sw)}$ is observed. Further north at the station at 4° N 10° W (Fig. 3f), Koffi et al. (2010) suggested that this region follows a similar seasonal cycle as the station at 6° S 10° W, but that $pCO_{2\ (sw)}$ is ~30 µatm lower (Koffi et al., 2016). The interannual

variability in SA-FNN$_{NCP}$ $pCO_{2\ (sw)}$ clearly shows the influence of the equatorial upwelling at these stations, with latitudinal gradients in $pCO_{2\ (sw)}$ during the upwelling period (Lefèvre et al., 2016). By contrast, the SA-FNN$_{NO-BIO}$ indicated little influence from the equatorial upwelling, little interannual variability, and a depressed $pCO_{2\ (sw)}$ during the upwelling season. The two methods converge on the seasonal cycle at the remaining stations although significant offsets in the mean annual $pCO_{2\ (sw)}$ remain. The station at 35° S 18° W (Fig. 3c) has consistently been implied as a sink for $CO_2$. Lencina-Avila et al.

(2016) showed the region to have $pCO_{2\ (sw)}$ at 340 µatm and to be a sink for $CO_2$ between October to December. Similarly, Kitidis et al. (2017) implied that the region is a sink for $CO_2$ during March to April. The region has depressed $pCO_{2\ (sw)}$ due to high biological activity that originates from the Patagonian shelf and the South Subtropical Convergence Zone. The station at 45° S 50° W (Fig. 3d), has also been implied as a strong, but highly variable sink, where $pCO_{2\ (sw)}$ can be between ~280 µatm and ~380 µatm during austral spring, and is constant at ~310 µatm during austral autumn (Kitidis et al., 2017).

The SA-FNN$_{NCP}$ and SA-FNN$_{NO-BIO}$ methods reproduced the seasonal variability in the $pCO_{2\ (sw)}$ at these two stations accurately, but only the SA-FNN$_{NCP}$ captures the magnitude of the depressed $pCO_{2\ (sw)}$ at 45° S. Within the southern Benguela upwelling system, $pCO_{2\ (sw)}$ at the station 33° S 17° E (Fig. 3e) is influenced by gradients in the seasonal upwelling (Hutchings et al., 2009). Santana-Casiano et al. (2009) showed that $pCO_{2\ (sw)}$ varies from ~310 µatm in July to ~340 µatm in December and that the region is a $CO_2$ sink through the year. González-Dávila et al. (2009)

suggested however, that this $CO_2$ sink is highly variable during upwelling events, and that recently upwelled waters act as a source ($pCO_{2\ (sw)}$ > $pCO_{2\ (atm)}$) of $CO_2$ to the atmosphere (Gregor and Monteiro, 2013). The SA-FNN$_{NCP}$ and SA-FNN$_{NO-BIO}$ were able to reproduce the seasonal cycle, although the SA-FNN$_{NCP}$ correctly represented the seasonal differences in $pCO_{2\ (sw)}$ as reported by Santana-Casiano et al. (2009).

Overall, compared to the SA-FNN$_{NO-BIO}$ at these stations, the SA-FNN$_{NCP}$ better represents the seasonality and the

interannual variability of $pCO_{2\ (sw)}$ in the South Atlantic Ocean, especially in the Equatorial Atlantic. The SA-FNN$_{NO-BIO}$ and W2020 both displayed significant differences to the SA-FNN$_{NCP}$ at similar stations (Fig. 4) although their $pCO_{2\ (sw)}$ estimates were not always consistent. The SA-FNN method uses only *in situ* $pCO_{2\ (sw)}$ observations from the South Atlantic Ocean to train the FNNs. The W2020 uses global *in situ* $pCO_{2\ (sw)}$ observations to train FNNs for 16 provinces with similar seasonal cycles (Landschützer et al., 2014; Watson et al., 2020b). The W2020 will therefore be weighted to $pCO_{2\ (sw)}$ variability in

regions of relatively abundant *in situ* observations (i.e. Northern Hemisphere) and may not be fully representative of the South Atlantic Ocean. This would explain the SA-FNN$_{NO-BIO}$ and W2020 differences, when driven using the same input variables.

Comparing the SA-FNN$_{NCP}$ and SA-FNN$_{CHLA}$ there were two significant differences (Fig. 4). A difference in the seasonal cycle in the southern Benguela (Fig. 3e) was observed. Santana-Casiano et al. (2009) showed that the minima $pCO_{2\ (sw)}$ in





July and maxima in December, consistent with the SA-FNN$_{NCP}$ and SA-FNN$_{NPP}$ whereas the SA-FNN$_{CHL}$ estimated the opposite scenario. Lamont et al. (2014) reported Chl $a$ concentrations to remain consistent in May and October, but NPP rates were significantly higher in October, associated with increased surface PAR and enhanced upwelling. The disconnect between Chl $a$, NPP and NCP limits the ability of Chl $a$ to estimate pCO$_{2\,(sw)}$, which is highlighted by the failure of the SA-FNN$_{CHLA}$ to identify the seasonal pCO$_{2\,(sw)}$ cycle.

A Chl $a$ to NPP disconnect, due to light limitation caused by suspended sediments, has also been reported in the Amazon Plume (Smith and Demaster, 1996), where a significant offset between the SA-FNN$_{NCP}$ and SA-FNN$_{CHLA}$ was observed (Fig. 3g; Fig. 4). Lefèvre et al. (2017) reported pCO$_{2\,(sw)}$ values ranging from 400 $\pm$ ~10 µatm in January to ~240 $\pm$ ~70 µatm in May. Although, the SA-FNN$_{NCP}$ January estimates are consistent, the May estimates are higher than these *in situ* observations. These observations were made further north (6° N) where the turbidity within the plume has decreased

sufficiently for irradiance to elevate NPP rates (Smith and Demaster, 1996), which decrease pCO$_{2\,(sw)}$. Chl $a$ remains relatively consistent across the plume (not shown), suggesting a disconnect between Chl $a$ and NPP at 4° N 50° W which would lead to lower pCO$_{2\,(sw)}$ estimates by the SA-FNN$_{CHLA}$, where NPP rates are low due to light limitation (Chen et al., 2012; Smith and Demaster, 1996). Respiration would be elevated from the decomposition of riverine organic material reducing NCP further (Cooley et al., 2007; Jiang et al., 2019; Lefèvre et al., 2017). It is noted that the Amazon Plume is a

dynamic region with transient, localised biological and pCO$_{2\,(sw)}$ features (Cooley et al., 2007; Ibánhez et al., 2015; Lefèvre et al., 2017; Valerio et al., 2021) that maybe masked by the coarse resolution of estimates available using satellite data. The SA-FNN$_{NCP}$ however, agreed with *in situ* pCO$_{2\,(sw)}$ observations at 4° N 50° W where pCO$_{2\,(sw)}$ varied above and below pCO$_{2\,(atm)}$ (Valerio et al., 2021).

       Though the differences between the SA-FNN$_{NCP}$ and SA-FNN$_{CHLA}$ may appear small, the Amazon Plume and Benguela

Upwelling have a higher intensity in the CO$_2$ flux per unit area compared to the open ocean, illustrating a disproportionate contribution to the overall global CO$_2$ sink than their small areal coverage implies (Laruelle et al., 2014). The differences in the pCO$_{2\,(sw)}$ estimates result in a 22 Tg C yr$^{-1}$ alteration in the annual CO$_2$ flux for the South Atlantic Ocean (SA-FNN$_{NCP}$ = +14 Tg C yr$^{-1}$; SA-FNN$_{CHLA}$ = -9 Tg C yr$^{-1}$; Fig. 5). This unequivocally reinforces the use of NCP to improve basin scale estimates of pCO$_{2\,(sw)}$, especially in regions where Chl $a$, NPP and NCP become disconnected.

Recent assessments of the strength of the global oceanic CO$_2$ sink have been made using pCO$_{2\,(sw)}$ fields estimated using no biological parameters as input (Watson et al., 2020b). Our results indicate that the SA-FNN$_{NCP}$ more accurately represented the pCO$_{2\,(sw)}$ variability in the South Atlantic Ocean compared to the SA-FNN$_{NO-BIO}$. Estimating the South Atlantic Ocean net CO$_2$ flux with the SA-FNN$_{NCP}$ pCO$_{2\,(sw)}$ produced a 14 Tg C yr$^{-1}$ source compared to a 10 Tg C yr$^{-1}$ sink indicated by the SA-FNN$_{NO-BIO}$ (Fig. 5). The incremental inclusion of parameters to properly account for the biological signal starting with

Chl $a$ (-9 Tg C yr$^{-1}$) to NPP (-7 Tg C yr$^{-1}$) to NCP (+14 Tg C yr$^{-1}$) switched the South Atlantic Ocean from a CO$_2$ sink to a source, driven by differences in the pCO$_{2\,(sw)}$ estimates in regions that are biologically controlled, such as the Equatorial Atlantic.





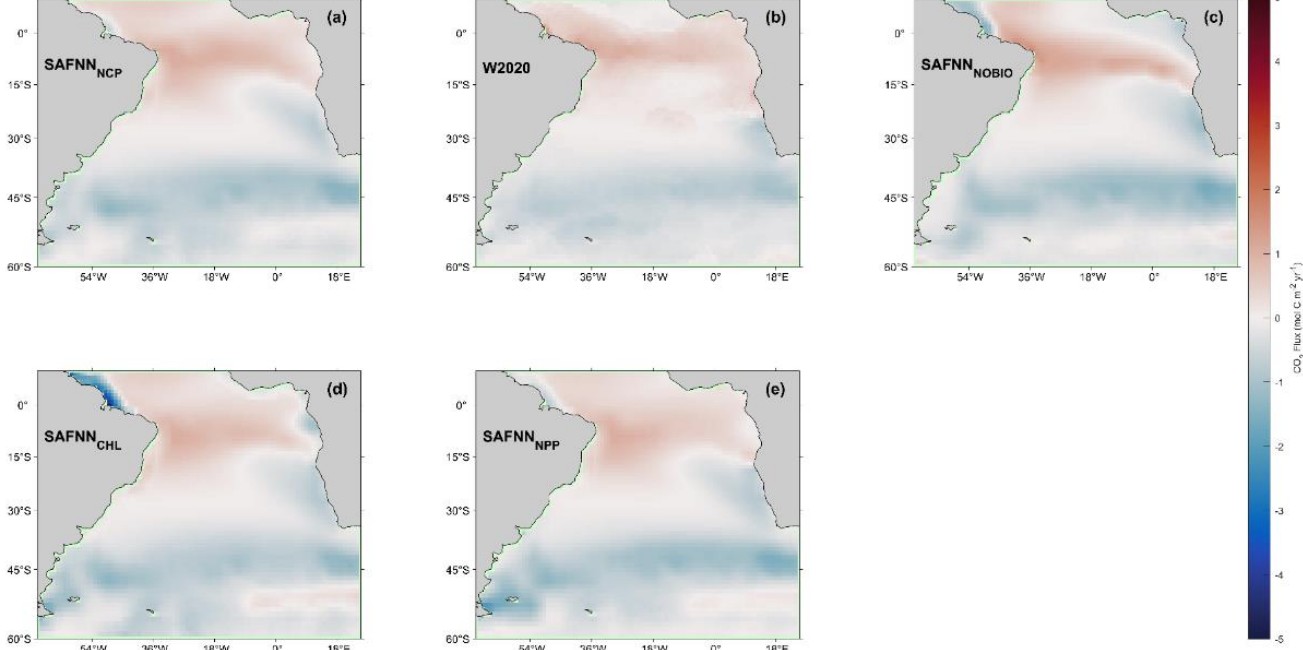

**Fig. 5: Long term average annual mean $CO_2$ flux for the South Atlantic Ocean, using $pCO_{2\,(sw)}$ estimates from (a) SA-FNN$_{NCP}$, (b) W2020 (Watson, et al., 2020a), (c) SA-FNN$_{NOBIO}$, (d) SA-FNN$_{CHLA}$ and (e) SA-FNN$_{NPP}$**

The W2020 identified the South Atlantic Ocean as a 15 Tg C yr$^{-1}$ source for $CO_2$ consistent with the SA-FNN$_{NCP}$ (Fig. 5). The SA-FNN$_{NCP}$ however, indicated the Equatorial Atlantic (10° N to 20° S) as a 20 Tg C yr$^{-1}$ stronger source and south of

20° S (20° S to 44° S) as a 20 Tg C yr$^{-1}$ stronger sink. These differences indicate that biologically induced variability in $pCO_{2\,(sw)}$ would not be captured by the W2020 and may reduce the variability in the global ocean $CO_2$ sink reinforcing the improvement that NCP provides.

## 5. Conclusions

In this paper, we compare using three biological proxies, Chl *a*, NPP or NCP available from earth observation data to train a

neural network scheme to estimate $pCO_{2\,(sw)}$. The results suggest that using NCP improved the estimation of $pCO_{2\,(sw)}$. The differences between satellite Chl *a*, NPP or NCP were initially small, but the use of a perturbation analysis to account for uncertainties in these parameters, showed that NCP has the greatest potential reduction in $pCO_{2\,(sw)}$ uncertainty of up to ~36 % of the RMSD, compared to a ~19 % reduction for Chl *a*. These results were verified using *in situ* observations from the Atlantic Meridional Transect, which resulted in a 25 % improvement in $pCO_{2\,(sw)}$ RMSD when the *in situ* NCP uncertainties





are reduced, compared to 7 % for Chl *a* and 13 % for NPP. Practical approaches to reduce the uncertainties in both the *in situ* and satellite NCP observations are discussed.

Monthly climatological estimates of $pCO_{2 (sw)}$ calculated using satellite NCP were compared with the NPP and the Chl *a* approaches and a baseline approach that does not use biological parameters, at 8 stations in the South Atlantic Ocean. The NCP approach significantly improved on the baseline approach at 4 stations in reconstructing the seasonal and interannual

variability, compared to *in situ* $pCO_{2 (sw)}$ observations. At the remaining 4 stations, differences were also observed although these were not statistically significant. A significant difference between the NCP and NPP approaches occurred in the eastern Equatorial Atlantic, in the equatorial upwelling region. Significant differences between the NCP and Chl *a* approaches were also observed in the Benguela upwelling and Amazon Plume, where $pCO_{2 (sw)}$ from Chl *a* suggested that photosynthetic rates were not solely controlled by Chl *a*. Using $pCO_{2 (sw)}$ estimated from NCP identified the South Atlantic Ocean as a net source

of $CO_2$, whereas methods that only include physical controls have indicated it to be a small sink for $CO_2$. Sequentially using firstly Chl *a* to estimate $pCO_{2 (sw)}$, then NPP incrementally reduced the South Atlantic $CO_2$ sink and finally using NCP switched the area to a source of $CO_2$. These results indicate that in regions where biological activity is important in controlling the variability in $pCO_{2 (sw)}$, the use of NCP available from satellite data is important for quantifying the ocean carbon pump, and for providing data in areas that are sparsely covered by observations such as the Southern Ocean.




## Appendices

### Appendix A - Feed forward neural network training and perturbation analysis

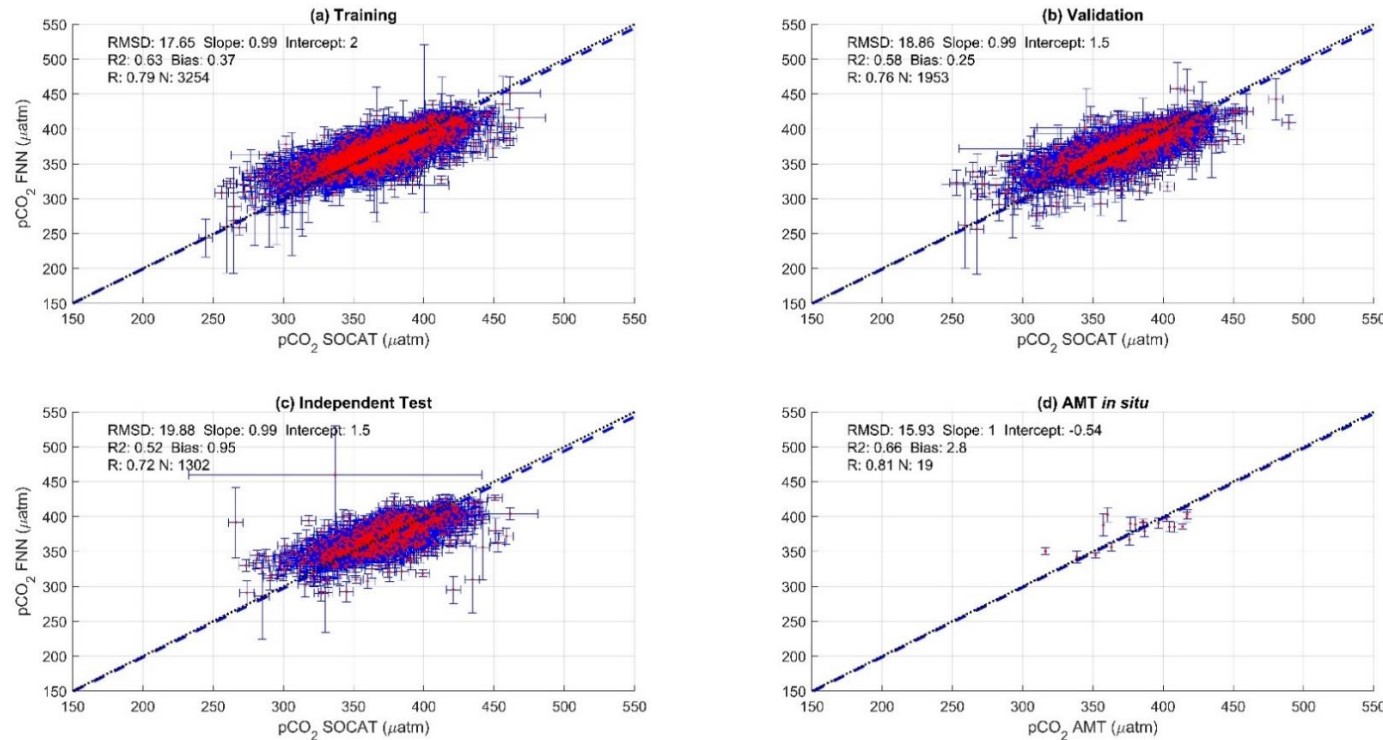

**Fig. A1: Scatter plots showing the combined performance of the 8 feed forward neural networks trained using chlorophyll a for 4 separate training and validation datasets; (a) Training, (b) Validation, (c) Independent Test and (d) Atlantic Meridional Transect (AMT) in situ. The blue dashed line is the Type II regression and the black dashed line is the 1:1 line. Horizontal errorbars indicate the uncertainty of the SOCATv2020 $pCO_{2\ (sw)}$. Vertical errorbars indicate the uncertainty attributed to the input parameter uncertainty propagated through the feed forward neural networks. The statistics within each plot are; Root Mean Square Difference (RMSD), Slope and Intercept of the Type II regression, Coefficient of Determination (R2), Pearson's Correlation Coefficient (R), Bias and number of samples (N).**





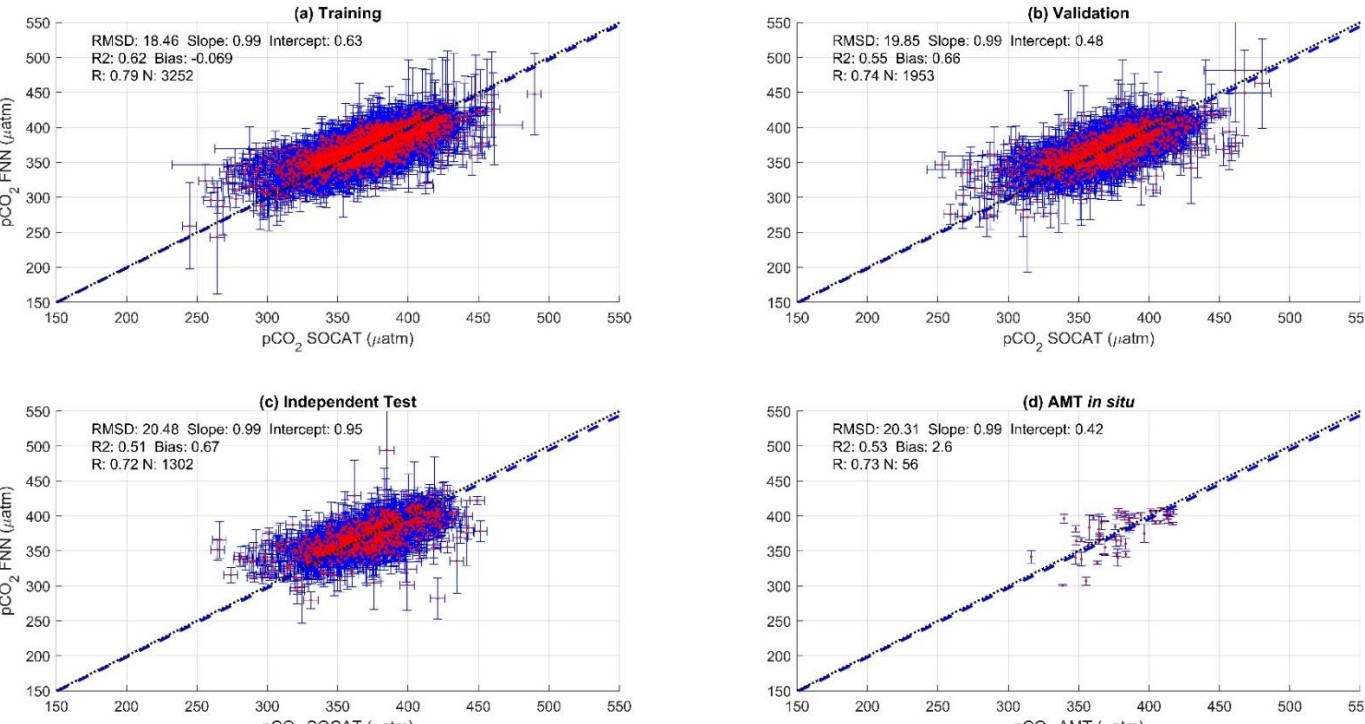

**Fig. A2: Scatter plots showing the combined performance of the 8 feed forward neural networks trained using net primary production for 4 separate training and validation datasets; (a) Training, (b) Validation, (c) Independent Test and (d) Atlantic Meridional Transect (AMT) in situ. The blue dashed line is the Type II regression and the black dashed line is the 1:1 line. Horizontal errorbars indicate the uncertainty of the SOCATv2020 $pCO_{2\ (sw)}$. Vertical errorbars indicate the resulting uncertainty attributed to the input parameter uncertainty propagated through the feed forward neural networks. The statistics within each plot are; Root Mean Square Difference (RMSD), Slope and Intercept of the Type II regression, Coefficient of Determination (R2), Pearson's Correlation Coefficient (R), Bias and number of samples (N).**






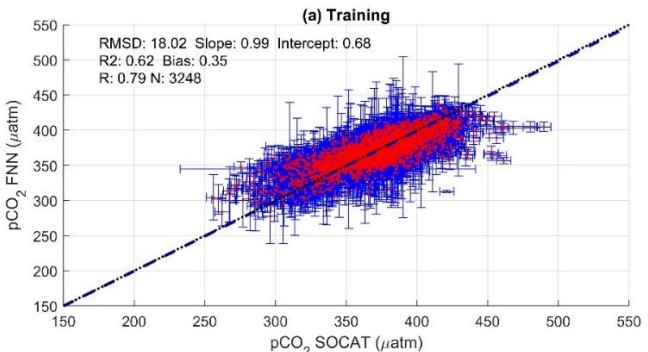
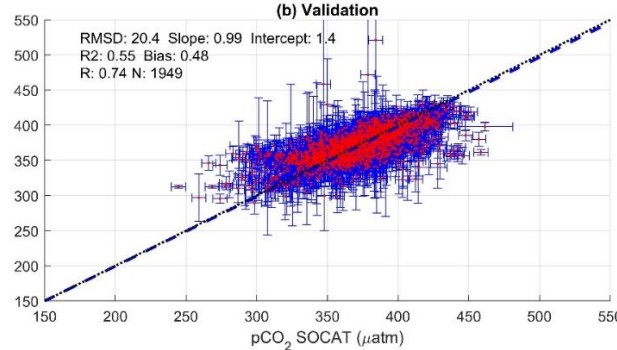

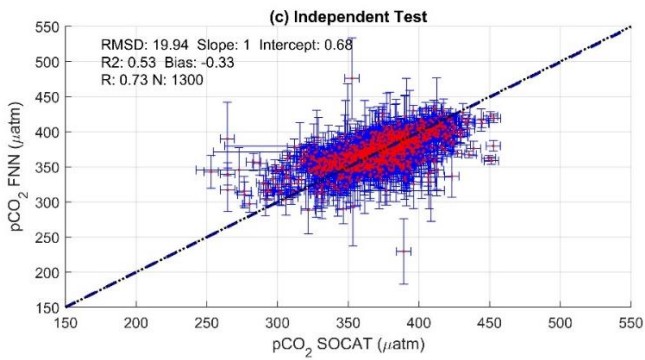


**Fig. A3: Scatter plots showing the combined performance of the 8 feed forward neural networks trained using no biological parameters for 3 separate training and validation datasets; (a) Training, (b) Validation and (c) Independent Test. The blue dashed line is the Type II regression and the black dashed line is the 1:1 line. Horizontal errorbars indicate the uncertainty of the SOCATv2020 pCO$_{2\ (sw)}$. Vertical errorbars indicate the resulting uncertainty attributed to the input parameter uncertainty**
**propagated through the feed forward neural networks. The statistics within each plot are; Root Mean Square Difference (RMSD), Slope and Intercept of the Type II regression, Coefficient of Determination (R2), Pearson's Correlation Coefficient (R), Bias and number of samples (N).**




**Table A1: The percentage reduction in Root Mean Square Difference (RMSD) attributable to the uncertainties in the input parameter for each training and validation datasets determined from a perturbation analysis as described in Sect. 2.5.**

| | Parameter | Training | Validation | Independent Test | AMT *in situ* |
|---|---|---|---|---|---|
| **NCP** | ALL | 33 % | 42 % | 38 % | 28 % |
| | SST | 10 % | 12 % | 10 % | 0.5 % |
| | Net Community Production | 32 % | 40 % | 36 % | 25 % |
| | $pCO_{2\,(atm)}$ | 6 % | 7 % | 6 % | 9 % |
| **Net Primary Production** | ALL | 34 % | 40 % | 40 % | 17 % |
| | SST | 9 % | 10 % | 10 % | 0.4 % |
| | Net Primary Production | 31 % | 37 % | 36 % | 13 % |
| | $pCO_{2\,(atm)}$ | 6 % | 6 % | 6 % | 9 % |
| **Chlorophyll *a*** | ALL | 22 % | 26 % | 25 % | 29 % |
| | SST | 9 % | 10 % | 9 % | 0.4 % |
| | Chlorophyll *a* | 17 % | 21 % | 20 % | 7 % |
| | $pCO_{2\,(atm)}$ | 8 % | 9 % | 9 % | 16 % |

**Appendix B - Climatology comparison**

A monthly climatology was generated from the SA-FNN$_{NCP}$ monthly timeseries (Fig. B1), referenced to the year 2010, assuming an atmospheric $CO_2$ increase of 1.5 µatm yr$^{-1}$ (Takahashi et al., 2009; Zeng et al., 2014). The standard deviation of the monthly climatology was computed, as an indication of the interannual variations in the climatology. The ability of the SA-FNN$_{NCP}$ to estimate the spatial distribution of $pCO_{2\,(sw)}$ was compared to two methods.

Firstly, the SA-FNN$_{NCP}$ climatology was compared to the climatology from Woolf et al. (2019), produced following the
statistical 'ordinary block kriging' approach described in Goddijn-Murphy et al. (2015), using the SOCATv4 reanalysed data. The method provides an interpolation uncertainty where in regions of sparse data this becomes larger. Fig. B2 shows the methods produce similar climatological $pCO_{2\,(sw)}$ values for the South Atlantic Ocean, with some clear differences along the African coastline, and equatorial region.

Secondly, the SA-FNN$_{NCP}$ was compared to a climatology calculated from the 'standard method', a Self Organising Map
Feed Forward Neural Network presented in Watson et al. (2020b; W2020). Fig. B3 shows the methods produce similar



climatological pCO$_{2\ (sw)}$ values for the South Atlantic Ocean, however, clear differences in the Equatorial region occur across all months. In the central South Atlantic Ocean, artefacts form the self organising map can be seen during January and February.

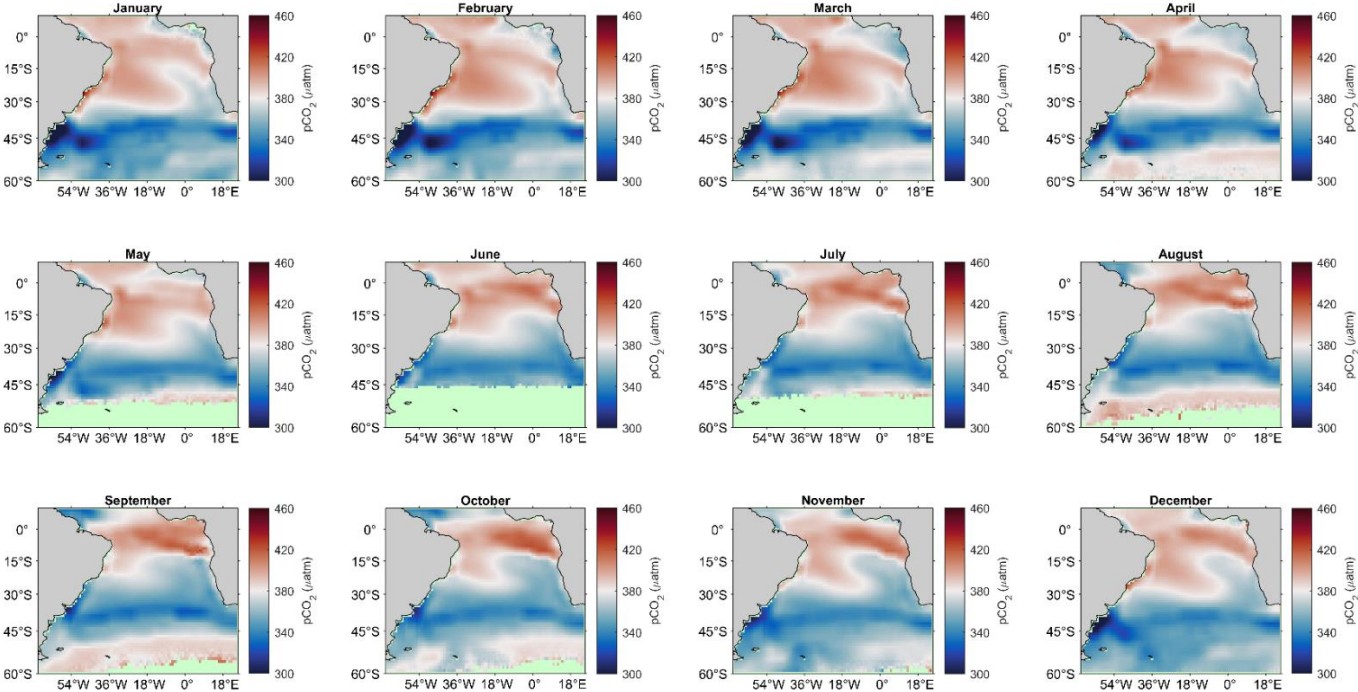

**Fig. B1: Monthly climatologies of pCO$_2$ $_{(sw)}$ between July 2002 and December 2018 estimated by the SA-FNN$_{NCP}$ approach referenced to 2010. The atmospheric CO$_2$ increase was set as 1.5 µatm yr$^{-1}$. The colour scale is centred on the atmospheric concentration for 2010 (~380 µatm). Red shaded areas indicate oversaturated regions, and blue shaded areas indicate under saturated regions. Light green areas indicate where no input data to compute pCO$_2$ $_{(sw)}$ are available.**





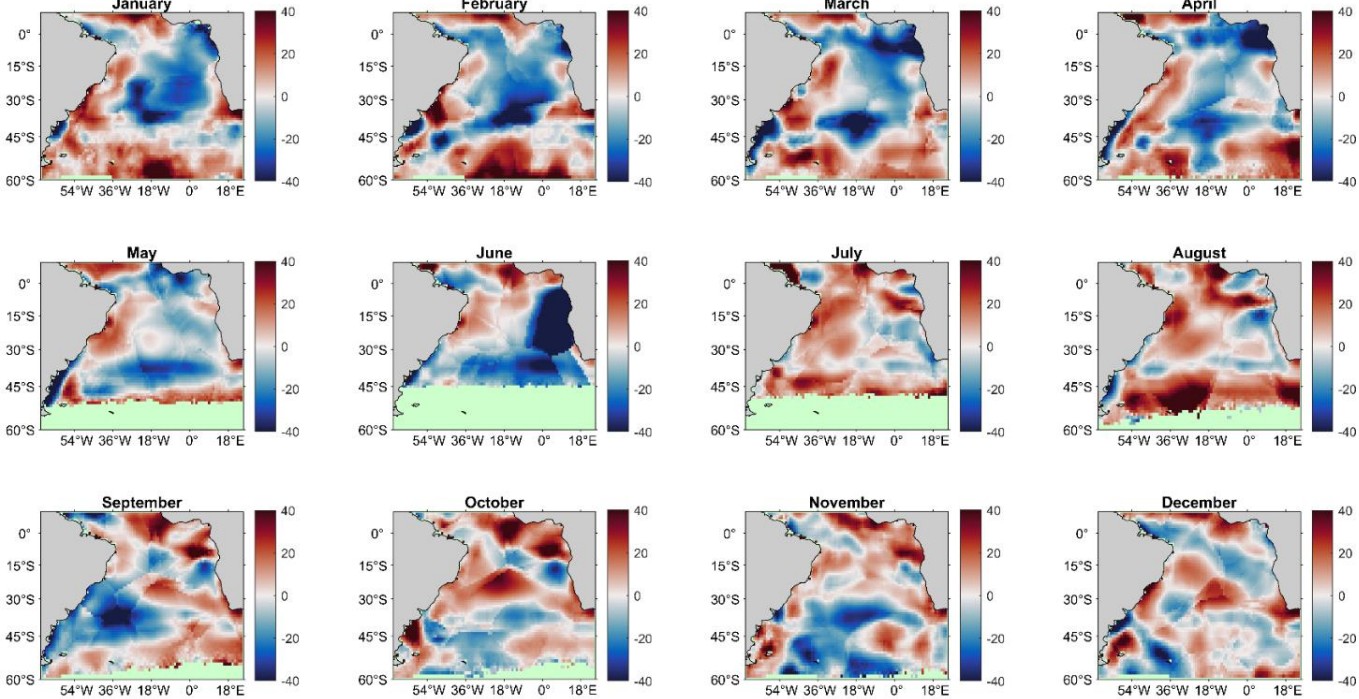

**Fig. B2: Monthly comparison between pCO₂ (sw) climatology estimated by the SA-FNN_NCP and Woolf et al (2019) climatology referenced to 2010 (SA-FNN_NCP pCO₂ − Woolf pCO₂). Red (Blue) shades indicate regions where SA-FNN is greater (less) than the Woolf climatology.**



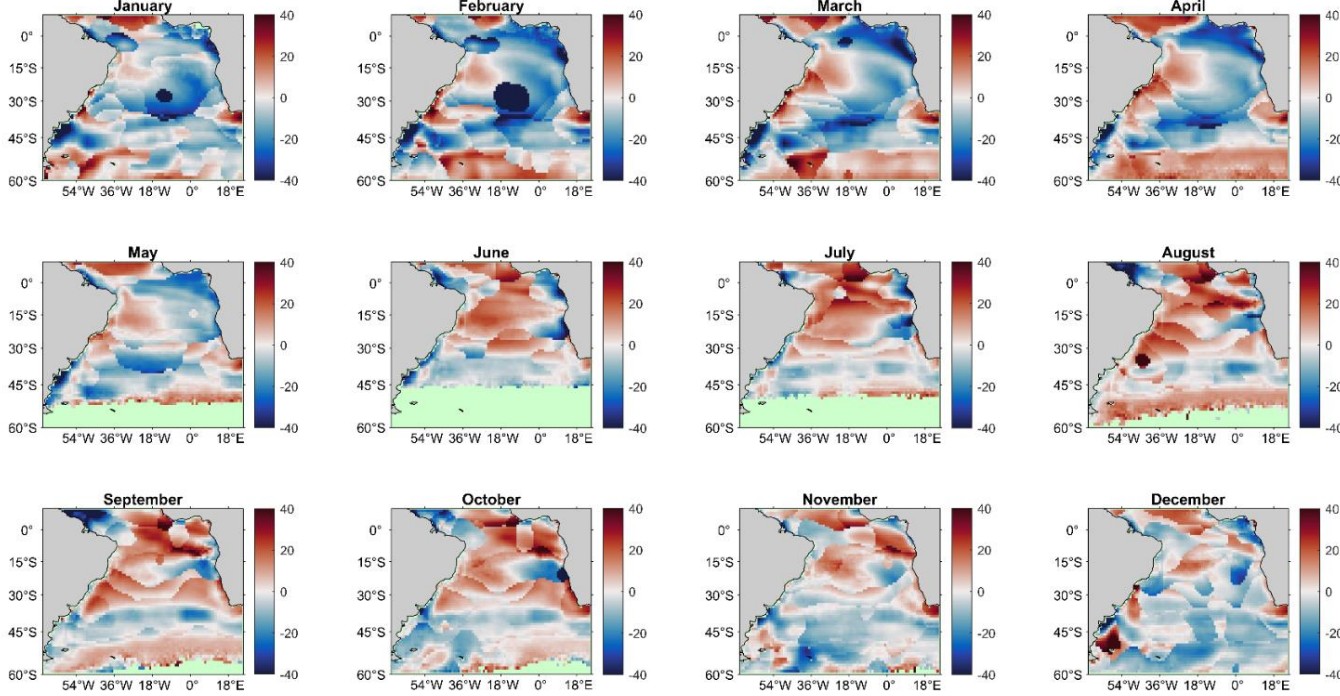

**Fig. B3: Monthly comparison between pCO$_{2\ (sw)}$ climatologies estimated by the SA-FNN$_{NCP}$ and W2020 (Watson et al, 2020)**
**climatology referenced to 2010 (SA-FNN$_{NCP}$ pCO$_2$ − W2020 pCO$_2$). Red (Blue) shades indicate regions where SA-FNN$_{NCP}$ is greater (less) than the W2020 climatology.**

**Data Availability**

Moderate Resolution Imaging Spectroradiometer on Aqua (MODIS-A) estimates of chlorophyll-a, photosynthetically active
radiation and sea surface temperature are available from the National Aeronautics Space Administration (NASA) ocean
colour website (https://oceancolor.gsfc.nasa.gov/). Modelled sea surface salinity and mixed layer depth from the Copernicus
Marine Environment Modelling Service global ocean physics reanalysis product (GLORYS12V1) are available from
https://resources.marine.copernicus.eu/. ERA5 monthly reanalysis wind speeds are available from the Copernicus Climate
Data Store (https://cds.climate.copernicus.eu/) pCO$_{2\ (atm)}$ data are available from v5.5 of the global estimates of pCO$_{2\ (sw)}$
dataset (Landschützer et al., 2016, 2017). *In situ* observations of fCO$_{2\ (sw)}$ from v2020 of the Surface Ocean Carbon Atlas
(SOCAT) are available from https://www.socat.info/index.php/data-access/. *In situ* Atlantic Meridional Transect data can be
obtained from the British Oceanographic Data Centre (https://www.bodc.ac.uk/). pCO$_{2\ (sw)}$ estimates from the W2020 are



available from Watson et al. (2020b). $pCO_{2\,(sw)}$ estimates generated by the SA-FNN$_{NCP}$, SA-FNN$_{NPP}$, SA-FNN$_{CHLA}$ and SA-FNN$_{NO\text{-}BIO}$ are available from Pangaea (Ford et al., submitted).


## Author Contribution

DF, GT, JS and VK conceived and directed the research. DF developed the code and prepared the manuscript. GT, JS and VK provided comments that shaped the final manuscript.

## Competing Interests

The authors declare that they have no conflict of interest.

## Acknowledgements

DF was supported by a NERC GW4+ Doctoral Training Partnership studentship from the UK Natural Environment Research Council (NERC; NE/L002434/1). GT and VK were supported by the AMT4OceanSatFlux (4000125730/18/NL/FF/gp) contract from the European Space Agency and NERC National Capability funding to Plymouth Marine Laboratory for the

Atlantic Meridional Transect (CLASS-AMT).

We would like to thank the captain and crew of *RRS Discovery, RRS James Clark Ross* and *RRS James Cook* for conducting the Atlantic Meridional Transects (AMT). We also thank the Natural Environment Research Council Earth Observation Data Acquisition and Analysis Service (NEODAAS) for use of the Linux cluster to process the MODIS-A satellite imagery. The Surface Ocean $CO_2$ Atlas (SOCAT) is an international effort, endorsed by the International Ocean Carbon Coordination

Project (IOCCP), the Surface Ocean Lower Atmosphere Study (SOLAS) and the Integrated Marine Biosphere Research (IMBeR) program, to deliver a uniformly quality-controlled surface ocean $CO_2$ database. The many researchers and funding agencies responsible for the collection of data and quality control are thanked for their contributions to SOCAT. The AMT is funded by NERC through its National Capability Long-term Single Centre Science Programme, Climate Linked Atlantic Sector Science (NE/R015953/1). This study contributes to the international IMBeR project and is contribution number 366

of the AMT programme.

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
