# Peer review of "Derivation of seawater $pCO_2$ from net community production identifies the South Atlantic Ocean as a $CO_2$ source"

_Biogeosciences, 2021_

## Author Comment (AC1)

Research excellence supporting a sustainable ocean

5th October 2021.

Biogeosciences.

Dear Editor,

Thank you for your and the reviewer's comments on our manuscript entitled 'Derivation of seawater $pCO_2$ from net community production identifies the South Atlantic Ocean as a $CO_2$ source' by Ford, Tilstone, Shutler and Kitidis. We have addressed all of the comments raised and implemented the necessary changes to the updated version of the manuscript. We provide detailed responses to the comments below and hope you find these changes satisfactory.

We look forward to hearing from you
Yours sincerely,

[Figure]

Daniel Ford

Registered Office:
Prospect Place
The Hoe, Plymouth
PL1 3DH, UK

T +44 (0)1752 633100
E forinfo@pml.ac.uk
W www.pml.ac.uk
@PlymouthMarine

Patron: James Cameron
Registered charity number 1091222.
PML is a company limited by guarantee,
registered in England & Wales,
company number 4178503

[Figure]

**Response to Jonathan Sharp (Reviewer #1)**

**General Comments:**
Daniel Ford and coauthors use a feed-forward neural network (FNN) to estimate surface ocean partial pressure of $CO_2$ ($pCO_{2(sw)}$) in the South Atlantic Ocean. The authors test satellite chlorophyll a (Chl a), satellite-derived net primary production (NPP), and satellite-derived net community production (NCP) as biological predictors in the neural network to determine which produces the most accurate $pCO_{2(sw)}$. They find that using satellite-derived NCP as a predictor in the FNN scheme produces the most reliable $pCO_{2(sw)}$ reconstructions for the Amazon River plume and upwelling regions. They also show that, among the neural networks examined, the NCP-based FNN (SA-FNN$_{NCP}$) has the highest capacity for improved performance under scenarios of reduced uncertainty. For these reasons, the authors suggest that using satellite-derived NCP as a proxy for biological effects in surface reconstructions of $pCO_{2(sw)}$ may be desirable going forward. Finally, Ford et al. find that SA-FNN$_{NCP}$ indicates that the South Atlantic Ocean is a source of $CO_2$ to the atmosphere, whereas the FNNs with Chl a or NPP as a biological proxy or with no biological proxy all indicate that the South Atlantic Ocean is a $CO_2$ sink.
This manuscript fits well within the scope of Biogeosciences: it explores the implications of choosing different biological predictor variables in estimation schemes for sea surface $pCO_2$ and demonstrates the consequences of those choices for carbon cycling calculations. It is based on the very logical assumption that NCP, which captures all biological processes that modulate $CO_2$ concentrations in the surface ocean, should serve as a better biological predictor than Chl a for estimates of $pCO_{2(sw)}$. This work has the potential to shift the way in which studies of this nature are typically performed. That shift could result in better representations of sea surface $pCO_2$ in regions that are highly influenced by biological processes, regions that may contribute a disproportionately large fraction of global $CO_2$ flux across the air–sea interface.
In general, the manuscript is well-written and the figures and tables are effective in communicating the results. The manuscript addresses an important aspect of the global carbon cycle and is forward-thinking in its assessment of algorithm performance in response to reduced uncertainties. A couple concerns of mine, however, include the lack of quantitative or graphical support for the conclusion that SA-FNN$_{NCP}$ produces the best representations of $pCO_{2(sw)}$ compared to the other FNNs (section 4.2) and the shortage of further investigation into one of the manuscript's major conclusions: that SA-FNN$_{NCP}$ flips the South Atlantic from a $CO_2$ sink to a $CO_2$ source. More discussion of these concerns as well as some minor comments can be found in the following sections.

*Response: Thank you for the appraisal of our manuscript and the comments that you have provided which have improved the manuscript. Detailed responses to each comment are given below.*

**Specific Comments:**
1. *Performance of SA-FNN$_{NCP}$:*

I mainly would like to see some quantitative or graphical evidence supporting the assertion that SA-FNN$_{NCP}$ outperforms the other FNNs in the Amazon River plume and upwelling regions. Figure 3 shows differences between the mean climatologies given by different FNNs at different stations and Figure 4 shows that some of these differences are statistically significant (in comparison to SA-FNN$_{NCP}$), but neither says anything about the performance of any one FNN. That is left to the more

qualitative discussion in section 4.2 that compares general patterns in $pCO_{2(sw)}$ from previous studies to those indicated by the FNNs.

The points made in that qualitative discussion are compelling and certainly do appear to indicate superior performance of SA-FNN$_{NCP}$ in the Amazon River plume, Benguela upwelling system, and equatorial regions. However, following along with the discussion takes some effort from the reader, and a lot of flipping back and forth between the text and Figure 3. A new figure comparing FNN results to some $pCO_{2(sw)}$ observations or a brief presentation of some relevant statistics would be more compelling. In particular, for example, $pCO_{2(sw)}$ data from the moorings at 6° S 10° W and/or 8° N 38° W could be plotted along with the SA-FNNs results to demonstrate the superior performance of SA-FNN$_{NCP}$.

I think this area is especially important to improve upon given that the bulk error statistics (Figures 2, A1, and A2) indicate SA-FNN$_{NCP}$ to be the least accurate of the three FNNs that have biological predictors.

**Response:** *Thank you for your suggestions, which we agree with as they make perfect sense. We have now added graphical evidence to support the discussion, as you suggest. Fig. 3 has been updated to include the PIRATA buoy data available in the SOCATv2020 Flag E dataset for 8°N 38°W and 6°S 10°W. These data were reanalysed to a consistent temperature and depth dataset in the same process as the SOCATv2020 data, so that all data have been treated in a consistent manner. We have also added graphical representations of the literature values described in the text to make the discussion easier to follow. Fig. 3 now also includes two SA-FNN variants (SA-FNN$_{NO\text{-}BIO\text{-}1}$ and SA-FNN$_{NO\text{-}BIO\text{-}2}$) which have no biological parameters as input. The updated Fig. 3 with its caption is given below:*

[Figure]

*Fig. 3: Monthly climatologies of $pCO_{2\,(sw)}$ referenced to the year 2010 for the 8 stations marked in Fig. 1 from the SA-FNN$_{NCP}$, SA-FNN$_{NPP}$, SA-FNN$_{CHLA}$, SA-FNN$_{NO\text{-}BIO}$ and W2020 (Watson et al., 2020b). Light blue lines in Fig. 3a, b indicate the in situ $pCO_{2\,(sw)}$ observations from PIRATA buoys. The atmospheric $CO_2$ increase was set*

[Figure]

Registered Office:
Prospect Place
The Hoe, Plymouth
PL1 3DH, UK

T  +44 (0)1752 633100
E  forinfo@pml.ac.uk
W  www.pml.ac.uk
   @PlymouthMarine

Patron: James Cameron
Registered charity number 1091222.
PML is a company limited by guarantee,
registered in England & Wales,
company number 4178503

[Figure]

*as 1.5 μatm yr⁻¹. Black dashed line indicates the atmospheric pCO₂ (~380 μatm). Error bars indicate the 2 standard deviation of the climatology (~95% interval), where larger error bars indicate a larger interannual variability. Red circles indicate the literature values of pCO₂ (sw) described in section 4.2. Note the different y-axis limits in Fig. 3g and Fig. 3d.*

*The updated Fig. 3 shows estimates of pCO₂ (sw) climatologies from 5 SA-FNN variants and the W2020 at 8 stations in the South Atlantic Ocean. The figure also shows climatologies computed from the in situ PIRATA buoy observations (in Fig. 3 a, b,) and the literature values described in the text. We have not however, produced statistical comparisons between the neural network estimates and the PIRATA data, because at 8°N 38°W the seasonal cycle is not fully captured, and the data only covers 4 years with patchy temporal coverage, so the datasets are not statistically comparable. For 6°S 10°W the data does capture the full seasonal cycle, but this consists of 6 years of data weighted towards June-December. As the neural network estimates cover 16 years, a direct statistical analysis is not possible. The inclusion of these data are beneficial to highlight that the SA-FNN_NCP does improve on the pCO₂ (sw) estimates, but there are still some differences between December and April. These have now been discussed directly in section 4.2.*

*2.* **Sink to source transition:**

The change in the cumulative regional sink from -7 Tg C yr⁻¹ with the NPP-based FNN (SA-FNN_NPP) to +14 Tg C yr⁻¹with SA-FNN_NCP seems rather drastic, and I'm curious to know more about why such a significant change occurs. The reason is not obvious from Figure 5 alone. If indeed the transition occurs because high outgassing events in biologically-controlled regions with relatively limited geographic extent are captured by SA-FNN_NCP but not the other FNNs, as is suggested in lines 399–412, that point should be demonstrated and emphasized more explicitly.

This could perhaps be explored by breaking down the annual fluxes into different sub-regions (e.g., the biogeochemical provinces from Figure 1) and/or into average monthly fluxes to clearly show the spatial and/or temporal differences that lead to the significant discrepancy between SA-FNN_NPP and SA-FNN_NCP. This information could be presented in a table, figure, or even just in the body of the manuscript (like the geographic comparison in lines 419–420 between SA-FNN_NCP and the Watson et al. [2020] product).

***Response:*** *We have now updated the text and relevant figures to strengthen the comparison between the SA-FNN_NCP and the other methods through the addition of a bar chart to Fig. 5. The bar chart shows the mean annual CO₂ fluxes (Fig. 5f) for 5 regions; (1) the whole South Atlantic, (2) 10°N to 20°S, (3) 20°S to 44°S, and additional the (4) WTRA and (5) ETRA provinces. The results for the SA-FNN_NO-BIO-1 and SA-FNN_NO-BIO-2 appear very similar. For completeness, both have been included in the bar chart, but since they are so similar, only the spatial distribution of the fluxes for SA-FNN_NO-BIO-2 are included (Fig. 5c). The updated Fig. 5 is given below:*

Registered Office: Prospect Place The Hoe, Plymouth PL1 3DH, UK | T +44 (0)1752 633100 E forinfo@pml.ac.uk W www.pml.ac.uk @PlymouthMarine | Patron: James Cameron Registered charity number 1091222. PML is a company limited by guarantee, registered in England & Wales, company number 4178503

[Figure]

*The bar chart shows the increased outgassing in the Equatorial region observed by the SA-FNN$_{NCP}$ compared to the other SA-FNN variants, and the overall $CO_2$ source for the South Atlantic Ocean estimated by the SA-FNN$_{NCP}$. We have expanded the discussion of the SA-FNN$_{NCP}$ and SA-FNN$_{NPP}$ differences. This appears in section 4.2 and reads:" The incremental inclusion of parameters to account for the biological signal starting with Chl a (-9 Tg C yr$^{-1}$) then NPP (-7 Tg C yr$^{-1}$) then NCP (+14 Tg C yr$^{-1}$) switched the South Atlantic Ocean from a $CO_2$ sink to a source, which is driven by differences in the $pCO_{2\,(sw)}$ estimates in regions that are biologically controlled. This 21 Tg C yr$^{-1}$ difference between the SA-FNN$_{NCP}$ and SA-FNN$_{NPP}$ is due to additional outgassing in the Equatorial Atlantic provinces of the WTRA and ETRA (Fig 1a; Fig.5f). Compared to the in situ $pCO_{2\,(sw)}$ observations at the Equatorial stations (Fig. 3a, b), it is likely that the outgassing is still underestimated by the SA-FNN$_{NCP}$ but does improve these estimates within the upwelling season (June – September).".*

**Callbacks in Discussion section:**
In general, because the work is presented in separate Results and Discussion sections, I'd make sure to refer specifically to figures, tables, and statements from the Results section when commenting on them in the Discussion section. This will make it easier for the reader to follow what exactly is being discussed without having to determine for themselves where those results are presented. I mention a couple specific instances of this in the following section.

*Response: Thank you for this suggestion, which will help the reader to follow the logic. We have now included within the discussion the requested specific references to the relevant tables and figures. For example, please see throughout section 4.1 where we have now referred to Fig. 2, and Table 2 and 3.*

**Minor Comments and Technical Corrections**
Line 17: There shouldn't be a comma after $pCO_{2(sw)}$.

[Figure]

Registered Office:
Prospect Place
The Hoe, Plymouth
PL1 3DH, UK

T   +44 (0)1752 633100
E   forinfo@pml.ac.uk
W  www.pml.ac.uk
    @PlymouthMarine

Patron: James Cameron
Registered charity number 1091222.
PML is a company limited by guarantee,
registered in England & Wales,
company number 4178503

*Response:* *We have removed this comma.*

Lines 45–48: I'd split this sentence into two; there's a lot of information here and it's a bit difficult to follow as written.

*Response:* *We have now split this sentence to provide the information on autotrophic communities in the first sentence, and heterotrophic communities in the second sentence. These sentences now read as follows: "When NCP is positive, the plankton community is autotrophic which implies that there is a drawdown of $CO_2$ from seawater (since the plankton reduce the $CO_2$ in the water column). Where NCP is negative the community is heterotrophic implying a release of $CO_2$ into the ocean (as the plankton produce or release $CO_2$) which can then be released into the atmosphere (Jiang et al., 2019; Schloss et al., 2007).".*

Figure 1: Do the colors and/or symbols represent different things in this figure? If so, I would mention it in the caption. If not, they could all be the same color/symbol since they're labeled with letters anyway. Additionally, can you add the AMT stations or the transect lines to this figure?

*Response:* *The different colours and symbols have no additional meaning to the stations, and therefore we have now made these the same symbol and colour. Additionally, and as requested, we have added the cruise tracks for AMT's 20, 21, 22 and 23. The updated Fig. 1 is given below:*

[Figure]

[Figure]

Table 1: What is the reference associated with the estimated uncertainty of 1 uatm in atmospheric $pCO_2$? Also, is it correct that the other uncertainty estimates all come from Ford et al. (2021)?

*Response:* *We have now included the reference for the 1 μatm uncertainty in atmospheric $pCO_2$, as Takahashi et al. (2009). The use of Ford et al. (2021) for the remaining parameters is correct. Ford et al. (2021) performed an uncertainty analysis on the Moderate Resolution Imaging Spectroradiometer on Aqua (MODIS-A) estimates of chlorophyll-a, net primary production, net community production and sea surface temperatures for the South Atlantic Ocean which has provided all of the other uncertainty components.*

Line 189: I'd rephrase this as "A non-parametric Kruskal-Wallis was used to test for…"

Registered Office:
Prospect Place
The Hoe, Plymouth
PL1 3DH, UK

T  +44 (0)1752 633100
E  forinfo@pml.ac.uk
W  www.pml.ac.uk
    @PlymouthMarine

Patron: James Cameron
Registered charity number 1091222.
PML is a company limited by guarantee,
registered in England & Wales,
company number 4178503

*Response: Thank you for the suggested rephrasing. We have updated the sentence which now reads: "A non-parametric Kruskal-Wallis was used to test for significant ($\alpha < 0.05$) differences in the annual $pCO_{2\,(sw)}$, indicating an offset between the two tested climatologies.".*

Line 232: Should the accuracy here for SA-FNN$_{NCP}$ be 21.68 matm, like in line 235 and Figure 2?

*Response: Yes, that is correct. We have now corrected this typo.*

Line 238: Based on Table 2, should these numbers be 36%, 36%, and 20%?

*Response: The average reduction of the training, validation and independent datasets is described in the text. This average difference was 36% for NCP, 34% for NPP and 19% for Chl a. We have now clarified this in the manuscript, which now reads: "This showed that satellite NCP uncertainties lead to an on average 36 % reduction in $pCO_{2\,(sw)}$ RMSD, with NPP a 34 % reduction and Chl a a 19 % reduction across the three training and validation datasets.".*

Lines 259–260 (and elsewhere): Should be "minimum" instead of "minima" and "maximum" instead of "maxima", or remove the article "a". Minimum/maximum are singular whereas minima/maxima are plural.

*Response: Thank you for highlighting this. We have now corrected this throughout the manuscript.*

Line 283: Add the accuracy of the SA-FNN$_{NCP}$ after you mention it here, so it can be easily compared with Landschützer et al. (2013) and Landschützer et al. (2014).

*Response: We have now added the SA-FNN$_{NCP}$ accuracy to the sentence as suggested. This now reads: "The SA-FNN$_{NCP}$ had an overall accuracy (21.68 μatm) which is consistent with other approaches that have been developed for the Atlantic (22.83 μatm; Landschützer et al., 2013), and slightly lower than the published global result of 25.95 μatm (Landschützer et al., 2014).".*

Lines 285–286: You've already mentioned in the previous sentence that the SA-FNN$_{NCP}$ approach has a similar accuracy to other approaches in the literature, so I'd say here that training the SA-FNN with Chl a or NPP gave comparable broad-scale accuracy to training it with NCP.

*Response: We have updated this sentence which now reads: "Training the SA-FNN using Chl a or NPP showed comparable broad-scale accuracy to NCP"*

Line 295: "Satellite NCP is reliant on NPP as input" This has already been implied in line 288, so I'd remove the statement here or move it to the previous paragraph. The point is well made, it's just that the writing is a bit repetitive here.

*Response: We have removed this sentence and the paragraph as you suggest. It now reads: "To uncouple the Chl a, NPP and NCP estimates and their uncertainties, the perturbation analysis was also conducted on Atlantic Meridional Transect in situ observations."*

Registered Office:
Prospect Place
The Hoe, Plymouth
PL1 3DH, UK

T +44 (0)1752 633100
E forinfo@pml.ac.uk
W www.pml.ac.uk
@PlymouthMarine

Patron: James Cameron
Registered charity number 1091222.
PML is a company limited by guarantee,
registered in England & Wales,
company number 4178503

Line 298–299: "This showed that reducing in situ NCP uncertainties provided the greatest reduction in $pCO_{2(sw)}$ RMSD, which was three times the reduction achievable using Chl a." I'd make sure to refer the reader to Tables 2 and 3 so they're not searching for this result.

*Response: We have now added the reference to Tables 2 and 3.*

Lines 328–329: "The stations (Fig. 1) represent locations of previous studies into in situ $pCO_{2(sw)}$ variability in the South Atlantic Ocean and allow comparisons with literature values." This point should be made earlier, in the Methods section, perhaps around line 182.

*Response: As suggested, we have now moved this sentence into the methods. This now reads: "The stations (Fig. 1) are representative of locations in the South Atlantic Ocean in the literature that have reported the variability of in situ $pCO_{2\ (sw)}$. For each station, the monthly climatology of $pCO_{2\ (sw)}$, representing the average seasonal cycle of $pCO_{2\ (sw)}$, and the standard deviation of the climatology, as an indication of the interannual variability, were extracted from the five approaches.".*

Lines 338–339: "Valerio et al. (2021) indicated $pCO_{2(sw)}$ varied above and below $pCO_{2(atm)}$ at 4° N 50° W consistent with the SA-FNN$_{NCP}$." It looks to me in Figure 3 like $pCO_{2(sw)}$ from SA-FNN$_{NCP}$ remains almost exclusively below $pCO_{2(atm)}$ at this site?

*Response: We have corrected the sentence after rechecking the data presented in Valerio et al. (2021), which showed $pCO_{2\ (sw)}$ to remain at or below $pCO_{2\ (atm)}$. This now reads: "The SA-FNN$_{NCP}$ however, agreed with in situ $pCO_{2\ (sw)}$ observations at 4° N 50° W where $pCO_{2\ (sw)}$ varied at or below $pCO_{2\ (atm)}$ (Valerio et al., 2021).".*

Line 396: Should be "may be" instead of "maybe"

*Response: This has now been corrected.*

Registered Office:
Prospect Place
The Hoe, Plymouth
PL1 3DH, UK

T  +44 (0)1752 633100
E  forinfo@pml.ac.uk
W  www.pml.ac.uk
   @PlymouthMarine

Patron: James Cameron
Registered charity number 1091222.
PML is a company limited by guarantee,
registered in England & Wales,
company number 4178503

**Response to Anonymous Reviewer (Reviewer #2)**

This manuscript explored the importance of biological factors in modeling the seawater pCO2 via the inclusion of different biological proxies (chl-a, NPP and NCP) in the model construction. The results reveal a similar model performance with the inclusion of different biological components but in some cases, projected pco2 with the inclusion of NCP as a predictor works better. Overall, the paper emphasizes the significance of biological activity in controlling the magnitude and temporospatial pattern of sweater pCO2 and underscores the urgency to further improve the accuracy in NCP prediction. However, I have some major concerns regarding the model test and data display.

*Response: Thank you for the appraisal of the manuscript. We have now addressed all of your concerns which have undoubtedly improved the work.*

Major comments:
1) Lack of comprehensive evaluation of multiple NPP and NCP algorithms. The key goal of your study is to elucidate the role of biological proxies in modeling seawater. Also, the author concluded the accuracy of NCP simulation is critical. If so, it is quite important to test the with the inclusion of commonly-used NPP and NCP algorithms (i.e, CbPM and VGPM-based NPP, NCP models by Li & Cassar, 2016; carbon export model by Laws et al., 2011 or (Henson et al., 2011)), rather than a single model output. Given that the focus is on the surface pCO2, I would say NCP model developed by Li and Cassar seems a better fit than model by Tilstone (2015) as the former provides mixed-layer integration of NCP rates and later is corresponding to euphotic-zone integration. At the very least, the author needs to clarify the reason why they chose the present NPP and NCP models (i.e., they are tuned by the Atlantic dataset and therefore they are supposed to do the best job in your study region). But I am still looking forward to seeing if other NPP and NCP products can further improve pco2 simulations.

*Response: The algorithms for Chl a, NPP and NCP are based on the work of Ford et al. (2021) which included an evaluation of the uncertainties (accuracy and bias) of the algorithms for the South Atlantic Ocean. In the Ford et al (2021) paper, five chl-a, three NPP and four NCP algorithms were evaluated, and the analysis is the first to account for both satellite and in situ uncertainties.*
*The NPP models assessed in Ford et al. (2021) were selected from models that had accurate performance in the Atlantic Ocean in previous studies (e.g. Campbell et al. 2002; Carr et al., 2006; Tilstone et al. 2009; Dogliotti et al., 2014; Lobanova et al. 2018), which included the VGPM. The NCP algorithms of Tilstone et al. (2015) were developed using bottle incubation data, and the algorithm used had the highest accuracy within the South Atlantic Ocean.*
*Thanks for the suggestion about the Li and Cassar (2016) model. This NCP model was tuned using $O_2/Ar$ ratio based measurements, where heterotrophic metabolic states (NCP<0) are not included. For the bottle incubations, which we used to validate the models, the NCP rates range from -50 to 50 mmol $O_2$ $m^{-2}$ $d^{-1}$. Since the $O_2/Ar$ NCP measurements are not comparable over this range, using algorithms developed using bottle incubations and those using $O_2/Ar$ NCP measurements are not directly comparable (please also see the discussions on this point within Duarte et al. 2013 and Williams et al. 2013). Therefore, evaluating the performance of the $pCO_{2\,(sw)}$ estimates using AMT data would not be applicable using the Li and Cassar (2016) approach. The export models of Laws et al. (2011) and Henson et al. (2011) do not derive NCP, but instead derive carbon export and are therefore not applicable.*

Registered Office: Prospect Place The Hoe, Plymouth PL1 3DH, UK

T +44 (0)1752 633100
E forinfo@pml.ac.uk
W www.pml.ac.uk
@PlymouthMarine

Patron: James Cameron
Registered charity number 1091222.
PML is a company limited by guarantee, registered in England & Wales, company number 4178503

We have now clarified the algorithm choices in the text as follows: *"These satellite algorithms were the most accurate for the South Atlantic Ocean in an algorithm inter-comparison which accounting for the uncertainties in both in situ, model and input data (Ford et al., 2021b).".*

2) Unfair comparison between FNNbio and FNNNO_bio: The author used the model output without the inclusion of biological predictors (FNNNO_bio) as a reference to evaluate the improvement of FNNbio. However, FNNNO_bio was trained with inclusion of more physical parameters such as MLD, SST and salinity whereas FNN_bio just included SST. To assure a fair comparison, it should keep other predictors consistency except for the inclusion or exclusion of biological parameters

**Response:** *We thank the Reviewer for their observation. To update the comparisons that we have conducted, we have now included a further SA-FNN$_{NO-BIO-1}$ which is trained with just pCO$_{2 (atm)}$ and SST.*
*For completeness, the original SA-FNN$_{NO-BIO}$, which is trained with pCO$_{2 (atm)}$, SST, SSS and MLD, is also included in the manuscript (now labelled as SA-FNN$_{NO-BIO-2}$). This method therefore allows us to make a direct comparison between the SA-FNN$_{NCP}$ and a SA-FNN trained with non-biological parameters consistent with the W2020 approach, but only using in situ pCO$_{2 (sw)}$ observations from the South Atlantic Ocean.*
*We have updated Figs. 3, 4 and 5 to include the SA-FNN$_{NO-BIO-1}$. The updated Fig 3 is given below:*

[Figure]

*This new Fig 3 shows that the SA-FNN$_{NO-BIO-1}$ and SA-FNN$_{NO-BIO-2}$ have similar pCO$_{2 (sw)}$ estimates at many of stations, but differences do occur within the Equatorial region. The pattern of the results is clearly shown in the updated Fig. 4, which is given below:*

Registered Office:
Prospect Place
The Hoe, Plymouth
PL1 3DH, UK

T  +44 (0)1752 633100
E  forinfo@pml.ac.uk
W  www.pml.ac.uk
    @PlymouthMarine

Patron: James Cameron
Registered charity number 1091222.
PML is a company limited by guarantee,
registered in England & Wales,
company number 4178503

[Figure]

*Fig. 4: Statistical comparison of the SA-FNN$_{NCP}$ with the W2020, SA-FNN$_{NO-BIO-1}$, SA-FNN$_{NO-BIO-2}$, SA-FNN$_{CHLA}$ and SA-FNN$_{NPP}$ climatologies, where yellow blocks indicate a significant difference ($\alpha = 0.05$). Seasonality indicates a difference in the seasonal cycle and offset indicates a difference between the mean pCO$_{2\ (sw)}$ of the climatologies.*

*The SA-FNN$_{NO-BIO-1}$ is also included in the mean annual CO$_2$ fluxes within the updated Fig. 5, which is given below:*

[Figure]

*Fig. 5f indicates that the SA-FNN$_{NO-BIO-1}$ and SA-FNN$_{NO-BIO-2}$ have very similar mean annual CO$_2$ flux values for the whole South Atlantic (10° N – 44° S). The two approaches do exhibit regional*

[Figure]

Registered Office:
Prospect Place
The Hoe, Plymouth
PL1 3DH, UK

T  +44 (0)1752 633100
E  forinfo@pml.ac.uk
W  www.pml.ac.uk
  @PlymouthMarine

Patron: James Cameron
Registered charity number 1091222.
PML is a company limited by guarantee,
registered in England & Wales,
company number 4178503

*differences however, at 10° N – 20° S and 20° S – 44° S. The result section 3.2 and discussion section 4.2 in the manuscript have now been modified to reflect the inclusion of the SA-FNN$_{NO-BIO-1}$.*

*For completeness and to address your comments, we also produced an SA-FNN trained using pCO$_{2\ (atm)}$, SST, salinity, MLD and NCP. This showed similar mean annual fluxes to the SA-FNN$_{NCP}$, and this variant and result is only included in the discussion section to avoid confusion with the other SA-FNN variants. Section 4.2 includes this paragraph which now states: 'A further SA-FNN trained with pCO$_{2\ (atm)}$, SST, salinity, mixed layer depth and NCP indicated a similar CO$_2$ source of 12 Tg C yr$^{-1}$ (data not shown) as the SA-FNN$_{NCP}$ for the South Atlantic Ocean, highlighting that additional physical parameters cannot fully account for the biological contribution to the variability in pCO$_{2\ (sw)}$. This further confirms the importance of using NCP within estimates of the global ocean CO$_2$ sink.'.*

3) The overstatement of the performance of FNNbio_NCP: In many places, the author argued that FNNbio_NC did the best among the models (i.e., Line 235, 360 and 365). However, it is hard to distinguish the visible improvement from the Figure. For example, I don't think the difference between 34% and 36% is significant enough to say NCP is much better than NPP given that uncertainty.

*Response: Thank you for pointing this out. We have now addressed this by either removing the statement (Line 235) or including further information in Fig. 3 to support the discussion at Lines 360 and 365. Please also see the responses to the minor comments below where this issue is also addressed.*

Minor comments:
1.p in pCO2 should be italic.

*Response: We have made this change throughout the manuscript.*

Line 15: it is hard to understand the method description regarding the "reducing uncertainty". Please clarify or say more specifically,

*Response: We have now clarified this sentence to make it clear that a perturbation analysis was used to explore the reduction in pCO$_{2\ (sw)}$ uncertainties. The sentence now reads: "A perturbation analysis explored the maximum reduction in pCO$_{2\ (sw)}$ uncertainties that could be achieved by reducing the uncertainties in the satellite biological parameters. This illustrated further improvement and greater differences for NCP compared to NPP or Chl a.".*

Line35 : clarify the full name of chl-a when it appears in the main text for the first time

*Response: We have now corrected this. Chlorophyll-a (Chl a) is now defined in the introduction.*

Line 60: do you have a specific reason why you focus on the Southern Atlantic instead of the entire Atlantic basin or larger scale. You need some statement herein and also add some brief introduction about the setting of your study region.

Registered Office:
Prospect Place
The Hoe, Plymouth
PL1 3DH, UK

T +44 (0)1752 633100
E forinfo@pml.ac.uk
W www.pml.ac.uk
@PlymouthMarine

Patron: James Cameron
Registered charity number 1091222.
PML is a company limited by guarantee,
registered in England & Wales,
company number 4178503

*Response: The South Atlantic Ocean was chosen as the study region because current pCO$_2$ $_{(sw)}$ estimates have shown a large divergence in this region (e.g. Rodenbeck et al., 2015). This is because there are limited data available in most of the Southern Hemisphere. As we describe in the introduction, the use of NCP may aid neural network extrapolation techniques by constraining the biological control on pCO$_2$ $_{(sw)}$. We also selected the South Atlantic Ocean due to the comprehensive uncertainty analysis that Ford et al. (2021) performed on the satellite parameters, which included the consideration of uncertainties in both the satellite and in situ data. These estimates provide a robust foundation for the investigation into the uncertainties that are introduced into pCO$_2$ $_{(sw)}$ by the input parameters.*

*We have now made this clearer in the introduction, alongside a description of the major features in the South Atlantic Ocean. The paragraph in the introduction now reads: "The South Atlantic Ocean is under sampled with limited pCO$_2$ $_{(sw)}$ observations (e.g. Fay and McKinley, 2013; Watson et al., 2020b). The region is varied and dynamic as it includes the seasonal Equatorial upwelling, high biological activity on the south-western (Dogliotti et al., 2014) and south-eastern shelves (Lamont et al., 2014), as well as the propagation of the Amazon Plume into the western Equatorial Atlantic (Ibánhez et al., 2015). This in conjunction with a comprehensive database of satellite observation-based data with associated uncertainties (Ford et al., 2021b) provides the potential to identify the improvement to pCO$_2$ $_{(sw)}$ estimates that could be made from using NCP."*

Figure 1. Please add the data distribution (i.e., SOCAT and AMT cruise) you used for construction and validation of the model. It should be useful to add the name of the sub-regions in the map so the reader can easily navigate what subregions you are talking about in the following part. Probably you need two figures to get two issues done. Add the citation for the region division on the legend.

*Response: Thank you for the suggestion. Fig. 1 has now been updated using two maps to address your comment. The first map shows the biogeochemical provinces, sub-region names, and the four Atlantic Meridional Transect cruise tracks as requested. The second map shows the frequency of the SOCAT dataset used in the training and validation of the SA-FNN. The updated Fig.1 is given below:*

[Figure]

[Figure]

*The Fig.1 caption has been updated to incorporate the changes, and now includes the citation of Longhurst et al. (1995) and Longhurst (1998). The legend now reads: "(a) Map of the 8 static biogeochemical provinces in the South Atlantic Ocean, following Longhurst et al. (1995) and Longhurst (1998). Markers and letters indicate the locations of timeseries extracted from Fig. 3. The four Atlantic Meridional Transect (AMT) cruise tracks are also overlaid (b) Map showing the spatial*

Registered Office:
Prospect Place
The Hoe, Plymouth
PL1 3DH, UK

T +44 (0)1752 633100
E forinfo@pml.ac.uk
W www.pml.ac.uk
@PlymouthMarine

Patron: James Cameron
Registered charity number 1091222.
PML is a company limited by guarantee,
registered in England & Wales,
company number 4178503

*distribution of the SOCATv2020 dataset used, where the data frequency is the number of available months of data within each 1° pixel. The province areas acronyms are: WTRA is Western Tropical Atlantic; ETRA is Eastern Equatorial Atlantic; SATL is South Atlantic Gyre; BRAZ is Brazilian current coastal; BENG is Benguela Current coastal upwelling; FKLD is Southwest Atlantic shelves; SSTC is South Subtropical Convergence; SANT is Sub Antarctic and ANTA is Antarctic.".*

Line 80: I would suggest changing pco2(atm) to pco2_air, which is more straightforward.

**Response:** *The use of $pCO_{2\,(atm)}$ is consistent with the nomenclature used for $pCO_{2\,(sw)}$ in this manuscript, and in previous manuscripts published in Biogeosciences (e.g. Landschützer et al., 2013; 2014). We have therefore left the notation as $pCO_{2\,(atm)}$. We hope that you find this acceptable.*

Table 1: does log10 means log10 (chl-a). It is very confusing. The error of 45 mmol O2 m-2 d-1 is very scaring because the typical range of NCP is from -50 and 50 mmol O2 m-2 d-1

**Response:** *The uncertainties in Chl a and NPP were assessed in $log_{10}$ space. We have now clarified that Chl a and NPP were $log_{10}$ transformed before input into the FNN, as follows: "Chl a and NPP estimates were $log_{10}$ transformed before input into the FNN, due to their respective uncertainties being determined in $log_{10}$ space.".*

Line 115: quite confused about what ropt means. Does it mean the optimal data numbers for the training dataset?

**Response:** *The $r_{opt}$ term indicates the optimal split in the input data to train and then validate the FNN, and this has been clarified in the methodology section 2.3.*

Line 125: The common practice for the data processing before training a neural network is to normalize the data to reduce the dynamics. Did you apply for it?

**Response:** *We applied no data processing to the input datasets (except for the $log_{10}$ transformation of Chl a and NPP). Our approach is similar to Landschutzer et al. (2013; 2014) who also applied no data processing, except for $log_{10}$ transformations of Chl a and Mixed Layer Depth. We have now included this detail within the methodology. This now reads (in section 2.3): "Chl a and NPP estimates were $log_{10}$ transformed before input into the FNN, due to their respective uncertainties which were determined in $log_{10}$ space.".*

Line 126: please provide more details about the structure of FNN. How many layers and nodes do you set?

**Response:** *The FNN consist of 1 input layer, 1 hidden layer, comprising of between 3-32 nodes depending on the results of the pre-training step described in the methodology and 1 output layer. We have now added this information into the methodology, which now reads (section 2.3): "The FNN consisted of 1 hidden layer with between 2 and 30 nodes depending on the pre-training step and 1 output layer."*

Line 145: Has pCO2 measurement in AMT cruise been included in the SOCAT dataset already?

[Figure]

Registered Office:
Prospect Place
The Hoe, Plymouth
PL1 3DH, UK

T +44 (0)1752 633100
E forinfo@pml.ac.uk
W www.pml.ac.uk
@PlymouthMarine

Patron: James Cameron
Registered charity number 1091222.
PML is a company limited by guarantee,
registered in England & Wales,
company number 4178503

*Response: Yes, the AMT $pCO_{2\,(sw)}$ observations are included within the SOCAT dataset. The data points ± 20 minutes surrounding the in situ Chl-a, NPP or NCP observations (N ≈ 200) were removed from the SOCATv2020 database. The removal of these data had minimal effect on the final gridded 1 degree monthly SOCAT observations used in training the SA-FNN. We have made this clear in the methodology describing the AMT datasets (Section 2.4). This reads: "These $pCO_{2\,(sw)}$ observations (N≈200) were removed from the SOCATv2020 dataset so that the Atlantic Meridional Transect data remained independent from the training and validation datasets."*

Table 2: you probably can bin Table 1 and Table 2 to make it easier for reading.

*Response: Table 1 and Table 2 are important for interpreting the results. Table 1 indicates the uncertainties on the parameters used in the SA-FNN, which are key for understanding the perturbation analysis and the propagation of these uncertainties through the SA-FNN. Table 2 provides results on the perturbation analysis assuming uncertainties can be reduced to 0 and is directly referred to within the results and discussion sections. To facilitate understanding the results and logic of the analyses, we have kept Table 1 and 2 in the paper. We hope that you understand the reasoning on this.*

Figure 2: the most useful information to evaluate the model performance is to look at the comparison with validation or independent dataset. I would suggest removing the figures of the training dataset and moving the validation results for models with inclusion of chla and NPP into the main texts.

*Response: The SA-FNN$_{NCP}$, SA-FNN$_{NPP}$ and SA-FNN$_{CHLA}$ have similar broad-scale accuracies, and the addition of the SA-FNN$_{NPP}$ and SA-FNN$_{CHLA}$ plots within the main manuscript would add no additional information. The inclusion of the RMSD and bias within the text provides the necessary information in a more concise form, to show these similarities in their accuracy.*

Figure 2: what does the color stand for? I don't think the blue line is useful. 1:1 line is enough. .

*Response: The colours were chosen to highlight the error bars (blue) and the data points (red). If only one colour was chosen it became unclear where the data points were when the error bars were added. We have now added this description to the figure caption, which reads: "The data points are highlighted in red to distinguish them from the error bars in blue."*
*The blue dashed line is the Type II weighted linear regression used in the validation of the SA-FNN outputs, as described in Ford et al. (2021). In cases where this line deviates from the 1:1, it would highlight the SA-FNN outputs had a systematic under/overestimation of $pCO_{2\,(sw)}$. This provides graphical evidence to the accuracy of the SA-FNN.*

Line 2165: I have a very hard time understanding how you derived so-called maximum reduction. To achieve this goal, as I understand, you need to set a wide range of noise in the predictors (i.e., from -500% to 500%) and then find out the largest errors in pCO2 compared to the white control (zero noise in predictors)?

*Response: What the reviewer describes is similar to the maximum reduction that is determined by the perturbation analysis, for the individual parameter reductions. The "noise" in the input parameter is*

[Figure]

Registered Office:
Prospect Place
The Hoe, Plymouth
PL1 3DH, UK

T  +44 (0)1752 633100
E  forinfo@pml.ac.uk
W  www.pml.ac.uk
   @PlymouthMarine

Patron: James Cameron
Registered charity number 1091222.
PML is a company limited by guarantee,
registered in England & Wales,
company number 4178503

set at the parameter's uncertainty (Table 1). The $pCO_{2\ (sw)}$ estimates from the neural network are calculated for 3 perturbations: input value + uncertainty, input value and input value – uncertainty. These three $pCO_{2\ (sw)}$ estimates are compared to the associated in situ $pCO_{2\ (sw)}$ observation, and the estimate with the smallest difference is selected. This is repeated for all the available coincident observations in the dataset, for example N = 1300 for the independent test dataset. The RMSD is then calculated between the selected perturbations and the in situ $pCO_{2\ (sw)}$. The percentage difference between this RMSD and the original RMSD (e.g. 21.68 µatm for independent test) is calculated, indicating the maximum reduction. We have now revised the methods section 2.5. to make this clearer.

Figure 4: how did you calculate the seasonality? Do you mean the amplitude of pco2? You can consider using Taylor diagrams to demonstrate the inter-model comparison, which provide more information.

**Response:** *The seasonal $pCO_{2\ (sw)}$ cycle at each station was estimated as the monthly climatology referenced to the year 2010. These were compared by calculating the Spearman Correlation coefficient (n=12) between the monthly climatology estimates from the SA-FNN$_{NCP}$ and sequentially the five other methods. Where the Spearman Correlation was insignificant, the seasonality was deemed significantly different. We have now included the phrase 'seasonality' within the methodology, which was previously missing.*

*The use of a Taylor Diagram would be useful in the context of having temporally complete in situ fields of $pCO_{2\ (sw)}$ to compare against. However, in the context of comparing multiple methods to the SA-FNN$_{NCP}$, without complete in situ data fields, there is no added benefit to using a Taylor Diagram, so it has not been added to the manuscript.*

Line 235: I don't think the difference between 34% and 36% is significant enough to say that the NCP is much better than NPP.

**Response:** *We have now removed the statement that reducing NCP uncertainties will have the largest impact on improving the $pCO_{2\ (sw)}$ at this line. The similar percentage reductions between satellite NCP and NPP is due to the coupling of their uncertainties, which is discussed in section 4.1. Performing the perturbation analysis on the in situ AMT data removed this coupling of uncertainties and showed NCP would have the greatest improvement, which is also discussed in section 4.1.*

Line 360 and Line 365: I had a hard time detecting significant improvement in SA-FNNNCP compared to the other simulations.

**Response:** *We have now updated Fig. 3 to include the literature values described in the text, to make it clearer about the variability discussed. The updated Fig. 3 is given below:*

Registered Office:
Prospect Place
The Hoe, Plymouth
PL1 3DH, UK

T   +44 (0)1752 633100
E   forinfo@pml.ac.uk
W   www.pml.ac.uk
    @PlymouthMarine

Patron: James Cameron
Registered charity number 1091222.
PML is a company limited by guarantee,
registered in England & Wales,
company number 4178503

Research excellence supporting a sustainable ocean

[Figure]

*In reference to the sections highlighted. Fig. 3d now shows the improvement in SA-FNN$_{NCP}$ pCO$_{2\ (sw)}$ compared to the literature values in austral autumn, where the SA-FNN$_{NO-BIO-1}$ indicates higher pCO$_2$ $_{(sw)}$ similar to the W2020. We now highlight in the discussion (section 4.2) that there is no significant difference between the SA-FNN$_{NCP}$ and the SA-FNN$_{CHLA}$ and SA-FNN$_{NPP}$ in Fig. 4.*
*Fig. 3e shows the literature values for the Benguela upwelling, which shows similarities between the SA-FNN$_{NCP}$ and SA-FNN$_{NO-BIO-1}$. We suggest that the SA-FNN$_{NCP}$ performs better and captures more of the variability due to upwelling through the larger interannual variability. We have further strengthened this section of the discussion by referring to the results of Arnone et al. (2017).*

Line 380: Can you find more straightforward evidence by examining the seasonality of chl-a, NPP and NCP products to support your argument regarding the disconnection?

***Response:*** *We have produced another figure for the appendix which displays the climatologies of Chl a, NPP and NCP at the 8 stations (Fig. C1) which is given below:*

Registered Office:
Prospect Place
The Hoe, Plymouth
PL1 3DH, UK

T  +44 (0)1752 633100
E  forinfo@pml.ac.uk
W  www.pml.ac.uk
@PlymouthMarine

Patron: James Cameron
Registered charity number 1091222.
PML is a company limited by guarantee,
registered in England & Wales,
company number 4178503

[Figure]

*The disconnect between Chl a and NPP reported by Lamont et al. (2014) can be seen in the satellite data, where NPP between April and October display similar values but Chl a is lower in October. The new Fig. C1 is referenced within section 4.2.*

Line 430: cite the corresponding Table and Figure when you talked about the result.

**Response:** *We have now corrected the omission of Table and Figure references throughout the manuscript, and at line 430.*

Line 440: it should be useful to plot a bar chart to display the annual co2 flux among the regions

**Response:** *We have now updated Fig. 5 to include a bar chart with the mean annual fluxes for the regions discussed in the text, and additionally for the WTRA and ETRA in line with another reviewer's suggestion. The updated figure is given below:*

Registered Office:
Prospect Place
The Hoe, Plymouth
PL1 3DH, UK

T  +44 (0)1752 633100
E  forinfo@pml.ac.uk
W  www.pml.ac.uk
    @PlymouthMarine

Patron: James Cameron
Registered charity number 1091222.
PML is a company limited by guarantee,
registered in England & Wales,
company number 4178503

[Figure]

Research excellence supporting a sustainable ocean

[Figure]

*The bar chart shows the overall CO₂ source estimated by the SA-FNN$_{NCP}$, mainly driven by increased outgassing in the Equatorial region. It also displays the similar overall CO₂ source estimate by the W2020, but the weaker sink between 20 °S – 44 °S and weaker source for 10 °N – 20 °S.*

Registered Office:
Prospect Place
The Hoe, Plymouth
PL1 3DH, UK

T   +44 (0)1752 633100
E   forinfo@pml.ac.uk
W   www.pml.ac.uk
    @PlymouthMarine

Patron: James Cameron
Registered charity number 1091222.
PML is a company limited by guarantee,
registered in England & Wales,
company number 4178503

---

## Author Response (AR1)

7th October 2021.

Biogeosciences.

Dear Peter Landschützer

Thank you for your and the reviewer's comments on our manuscript entitled 'Derivation of seawater  $pCO_2$  from net community production identifies the South Atlantic Ocean as a  $CO_2$  source' by Ford, Tilstone, Shutler and Kitidis. We have addressed all of the comments raised and implemented the necessary changes to the updated version of the manuscript. We provide detailed responses to the comments below and hope you find these changes satisfactory. In the responses below we refer to page and line numbers in the tracked changed document.

We look forward to hearing from you Yours sincerely,

Daniel Ford

- T +44 (0)1752 633100 E forinfo@pml.ac.uk W www.pml.ac.uk
- 9 @PlymouthMarine

**Response to Jonathan Sharp (Reviewer #1)**

**General Comments:**

Daniel Ford and coauthors use a feed-forward neural network (FNN) to estimate surface ocean partial pressure of CO2 (pCO2(sw)) in the South Atlantic Ocean. The authors test satellite chlorophyll a (Chl a), satellite-derived net primary production (NPP), and satellite-derived net community production (NCP) as biological predictors in the neural network to determine which produces the most accurate  $pCO_{2(sw)}$ . They find that using satellite-derived NCP as a predictor in the FNN scheme produces the most reliable  $pCO_{2(sw)}$  reconstructions for the Amazon River plume and upwelling regions. They also show that, among the neural networks examined, the NCP-based FNN (SA-FNNNCP) has the highest capacity for improved performance under scenarios of reduced uncertainty. For these reasons, the authors suggest that using satellite-derived NCP as a proxy for biological effects in surface reconstructions of pCO2(sw) may be desirable going forward. Finally, Ford et al. find that SA- $FNN_{NCP}$  indicates that the South Atlantic Ocean is a source of  $CO_2$  to the atmosphere, whereas the FNNs with Chl a or NPP as a biological proxy or with no biological proxy all indicate that the South Atlantic Ocean is a CO2 sink.

This manuscript fits well within the scope of Biogeosciences: it explores the implications of choosing different biological predictor variables in estimation schemes for sea surface pCO2 and demonstrates the consequences of those choices for carbon cycling calculations. It is based on the very logical assumption that NCP, which captures all biological processes that modulate CO2 concentrations in the surface ocean, should serve as a better biological predictor than Chl a for estimates of  $pCO_{2(sw)}$ . This work has the potential to shift the way in which studies of this nature are typically performed. That shift could result in better representations of sea surface  $pCO_2$  in regions that are highly influenced by biological processes, regions that may contribute a disproportionately large fraction of global CO2 flux across the air-sea interface.

In general, the manuscript is well-written and the figures and tables are effective in communicating the results. The manuscript addresses an important aspect of the global carbon cycle and is forwardthinking in its assessment of algorithm performance in response to reduced uncertainties. A couple concerns of mine, however, include the lack of quantitative or graphical support for the conclusion that SA-FNNNCP produces the best representations of  $pCO_{2(sw)}$  compared to the other FNNs (section 4.2) and the shortage of further investigation into one of the manuscript's major conclusions: that SA- $FNN_{NCP}$  flips the South Atlantic from a CO2 sink to a CO2 source. More discussion of these concerns as well as some minor comments can be found in the following sections.

**Response:** Thank you for the appraisal of our manuscript and the comments that you have provided which have improved the manuscript. Detailed responses to each comment are given below.

**Specific Comments:**

**1.** Performance of SA-FNNNCP:

I mainly would like to see some quantitative or graphical evidence supporting the assertion that SA-FNNNCP outperforms the other FNNs in the Amazon River plume and upwelling regions. Figure 3 shows differences between the mean climatologies given by different FNNs at different stations and Figure 4 shows that some of these differences are statistically significant (in comparison to SA- $FNN_{NCP}$ ), but neither says anything about the performance of any one FNN. That is left to the more

Registered Office: Prospect Place The Hoe, Plymouth PL1 3DH, UK

T +44 (0)1752 633100

E forinfo@pml.ac.uk

W www.pml.ac.uk

✓ @PlymouthMarine

qualitative discussion in section 4.2 that compares general patterns in  $pCO_{2(sw)}$  from previous studies to those indicated by the FNNs.

The points made in that qualitative discussion are compelling and certainly do appear to indicate superior performance of SA-FNNNCP in the Amazon River plume, Benguela upwelling system, and equatorial regions. However, following along with the discussion takes some effort from the reader, and a lot of flipping back and forth between the text and Figure 3. A new figure comparing FNN results to some  $pCO_{2(sw)}$  observations or a brief presentation of some relevant statistics would be more compelling. In particular, for example, pCO2(sw) data from the moorings at 6° S 10° W and/or 8° N 38° W could be plotted along with the SA-FNNs results to demonstrate the superior performance of SA-FNNNCP.

I think this area is especially important to improve upon given that the bulk error statistics (Figures 2, A1, and A2) indicate SA-FNNNCP to be the least accurate of the three FNNs that have biological predictors.

**Response:** Thank you for your suggestions, which we agree with as they make perfect sense. We have now added graphical evidence to support the discussion, as you suggest. Fig. 3 has been updated to include the PIRATA buoy data available in the SOCATv2020 Flag E dataset for 8°N 38°W and 6°S  $10^{\circ}$ W. These data were reanalysed to a consistent temperature and depth dataset in the same process as the SOCATv2020 data, so that all data have been treated in a consistent manner (detailed in section 2.1, Page 6 Lines 100 - 103). We have also added graphical representations of the literature values described in the text to make the discussion easier to follow. Fig. 3 now also includes two SA-FNN variants (SA-FNNNO-BIO-1 and SA-FNNNO-BIO-2) which have no biological parameters as input. The updated Fig. 3 (Page 15) with its caption is given below:

Fig. 3: Monthly climatologies of  $pCO_{2(sw)}$  referenced to the year 2010 for the 8 stations marked in Fig. 1 from the SA-FNNNCP, SA-FNNNPP, SA-FNNCHLA, SA-FNNNO-BIO and W2020 (Watson et al., 2020b). Light blue lines in

**Registered Office:** Prospect Place The Hoe, Plymouth PL1 3DH, UK

+44 (0)1752 633100

forinfo@pml.ac.uk E

W www.pml.ac.uk

@PlymouthMarine

Fig. 3a, b indicate the in situ  $pCO_{2(sw)}$  observations from PIRATA buoys. The atmospheric  $CO_2$  increase was set as 1.5 µatm yr-1. Black dashed line indicates the atmospheric  $pCO_2$  (~380 µatm). Error bars indicate the 2 standard deviation of the climatology (~95% interval), where larger error bars indicate a larger interannual variability. Red circles indicate the literature values of  $pCO_{2(sw)}$  described in section 4.2. Note the different yaxis limits in Fig. 3g and Fig. 3d.

The updated Fig. 3 shows estimates of  $pCO_{2 (sw)}$  climatologies from 5 SA-FNN variants and the W2020 at 8 stations in the South Atlantic Ocean. The figure also shows climatologies computed from the in situ PIRATA buoy observations (in Fig. 3 a, b,) and the literature values described in the text. We have not however, produced statistical comparisons between the neural network estimates and the PIRATA data, because at 8°N 38°W the seasonal cycle is not fully captured, and the data only covers 4 years with patchy temporal coverage, so the datasets are not statistically comparable. For 6°S 10°W the data does capture the full seasonal cycle, but this consists of 6 years of data weighted towards June-December. As the neural network estimates cover 16 years, a direct statistical analysis is not possible. The inclusion of these data are beneficial to highlight that the SA-FNNNCP does improve on the pCO2 (sw) estimates, but there are still some differences between December and April. These have now been discussed directly in section 4.2 (Page 19 Lines 380-382; Page 20 Lines 395-396).

**2. Sink to source transition:**

The change in the cumulative regional sink from -7 Tg C yr-1 with the NPP-based FNN (SA-FNNNPP) to +14 Tg C yr-1 with SA-FNNNCP seems rather drastic, and I'm curious to know more about why such a significant change occurs. The reason is not obvious from Figure 5 alone. If indeed the transition occurs because high outgassing events in biologically-controlled regions with relatively limited geographic extent are captured by SA-FNNNCP but not the other FNNs, as is suggested in lines 399–412, that point should be demonstrated and emphasized more explicitly.

This could perhaps be explored by breaking down the annual fluxes into different sub-regions (e.g., the biogeochemical provinces from Figure 1) and/or into average monthly fluxes to clearly show the spatial and/or temporal differences that lead to the significant discrepancy between SA-FNNNPP and SA-FNNNCP. This information could be presented in a table, figure, or even just in the body of the manuscript (like the geographic comparison in lines 419–420 between SA-FNNNCP and the Watson et al. [2020] product).

**Response:** We have now updated the text and relevant figures to strengthen the comparison between the SA-FNNNCP and the other methods through the addition of a bar chart to Fig. 5. The bar chart shows the mean annual CO2 fluxes (Fig. 5f) for 5 regions; (1) the whole South Atlantic, (2) 10°N to  $20^{\circ}$ S, (3)  $20^{\circ}$ S to  $44^{\circ}$ S, and additional the (4) WTRA and (5) ETRA provinces. The results for the SA-FNNNO-BIO-1 and SA-FNNNO-BIO-2 appear very similar. For completeness, both have been included in the bar chart, but since they are so similar, only the spatial distribution of the fluxes for SA-FNNNO-BIO-2 are included (Fig. 5c). The updated Fig. 5 (Page 23) is given below:

---

## Author Response (AR2)

9th November 2021.

Biogeosciences.

Dear Peter Landschützer,

Thank you for your and the reviewer's comments on our revised manuscript entitled 'Derivation of seawater $pCO_2$ from net community production identifies the South Atlantic Ocean as a $CO_2$ source' by Ford, Tilstone, Shutler and Kitidis. We have addressed all of the comments raised and implemented the necessary changes to the updated version of the manuscript. We provide detailed responses to your and Reviewer #1 comments below and hope you find these changes satisfactory. In the responses, we refer to page and line numbers in the tracked changed document.

We look forward to hearing from you
Yours sincerely,

[Figure]

Daniel Ford

Registered Office:    T  +44 (0)1752 633100    Patron: James Cameron
Prospect Place        E  forinfo@pml.ac.uk      Registered charity number 1091222.
The Hoe, Plymouth     W  www.pml.ac.uk          PML is a company limited by guarantee,
PL1 3DH, UK           🐦 @PlymouthMarine       registered in England & Wales,
                                               company number 4178503

**Response to Peter Landschützer (Editor)**

Dear authors,

I have now received the 2nd review of both referees and I agree with their judgement, that the manuscript has improved and is almost ready for publication. One referee has raised a number of additional comments that I would like you to consider. Therefore, I have decided that minor revisions are necessary before the manuscript can be considered for publication. However, once you have considered these final comments and resubmit your manuscript, I will proceed with my final decision (without consulting the referees again).

Best regards
Peter Landschützer

***Response:*** *Thank you for your decision and providing the reviewers comments. We have addressed the additional comments by the reviewers below, which have improved the manuscript further.*

**Response to anonymous reviewer (Reviewer #2)**

No additional comments provided

***Response:*** *Thank you for your second review of our manuscript.*

**Response to Jonathan Sharp (Reviewer #1)**

**General Comments:**

Daniel Ford and coauthors describe a study in which three different biological parameters — chlorophyll a (Chl a), net primary production (NPP), and net community production (NCP) — were tested as predictors in neural networks to estimate the partial pressure of CO2 in the surface ocean (pCO2(sw)) in the South Atlantic Ocean. Fields of pCO2(sw) generated by these three neural networks were compared to each other, as well as to fields generated by two additional neural networks that did not include biological predictors, a recently published global surface pCO2(sw) product (Watson et al., 2020), and in situ literature values of pCO2(sw). Also, a perturbation study was carried out to quantify the potential for improvements to pCO2(sw) predictions from each of the three neural networks with biological predictor parameters.

The authors conclude that the approach that includes NCP as a biological predictor provides the most accurate values of pCO2(sw) in equatorial upwelling regions and in the Amazon plume region. They demonstrate this result by comparing climatologies generated by the neural networks to in situ buoy measurements, values of pCO2(sw) reported in the literature, and climatologies generated by separate neural networks without biological predictors. They also conclude that the approach that includes NCP as a biological predictor has the greatest capacity for improvement to its performance as uncertainties are reduced.

Registered Office:
Prospect Place
The Hoe, Plymouth
PL1 3DH, UK

T  +44 (0)1752 633100
E  forinfo@pml.ac.uk
W  www.pml.ac.uk
  @PlymouthMarine

Patron: James Cameron
Registered charity number 1091222.
PML is a company limited by guarantee,
registered in England & Wales,
company number 4178503

[Figure]

The authors have responded well to the reviewers' comments, resulting in improvements to the presentation and discussion of their results. The modifications made to Figs. 1, 3, and 5 are especially helpful to the manuscript. I do support the publication of this work, as the implications are both important and interesting. Nevertheless, additional comments and editorial corrections are listed in the following section, which I hope will lead to further improvement to the manuscript.

*Response: Thank you for your second appraisal of the manuscript, and the additional comments provided which have improved the manuscript further. Responses to the specific comments are given below.*

**Specific Comments and Technical Corrections:**

Lines 9–10: Recommend revising to "As a part of this process…"

*Response: We have revised the start of this sentence as suggested on Page 1, Lines 9-10.*

Line 14 (and elsewhere): Recommend revising to "…which biological proxy produces the most accurate fields of pCO2(sw)."

*Response: We have revised this as suggested at Page 1, Line 14, and revised text at Page 3, Line 69 and Page 20, Line 477-478.*

Line 18: Add missing period after "parameters"

*Response: We have added the missing period on Page 1, Line 18.*

Line 20: Recommend revising to "…this region appears to be a sink for CO2"

*Response: This has been revised as suggested; see Page 1, Line 20.*

Line 45: Recommend revising to "Where NCP is positive…" to match the structure of the following sentence.

*Response: This has been revised as suggested; see Page 2, Lines 46-48.*

Line 64: Recommend revising to "This dynamic biogeochemical variability in conjunction with…" or something more descriptive than just "This"

*Response: We have revised the start of this sentence as suggested; see Page 3, Lines 65-67.*

Line 69: eliminate errant "a" between "alongside" and "two"

*Response: We have removed the "a"; see Page 3, Line 70.*

[Figure]

Registered Office:
Prospect Place
The Hoe, Plymouth
PL1 3DH, UK

T  +44 (0)1752 633100
E  forinfo@pml.ac.uk
W  www.pml.ac.uk
@PlymouthMarine

Patron: James Cameron
Registered charity number 1091222.
PML is a company limited by guarantee,
registered in England & Wales,
company number 4178503

Research excellence supporting a sustainable ocean

Lines 97–100: This paragraph seems unnecessary until reading in section 2.6 that the PIRATA buoy data are flagged E. I'd either mention the PIRATA data here, or just remove this paragraph. There is no mention of dataset quality flags in the preceding paragraph, so there is not necessarily a reason for the reader to assume that flag E data weren't also downloaded along with the core SOCAT data.

*Response: We agree and have now removed this paragraph; see Page 4, Lines 98 – 101.*

Line 115: Recommend revising to "These satellite algorithms were shown to be the most accurate…"

*Response: We have revised the sentence as suggested on Pages 4-5, Lines 116 - 117.*

Line 116: Change "accounting" to "accounted"

*Response: We have changed accounting to accounted on Page 5 Line 117.*

Lines 151–158: Although it is explained here, I was initially confused as to exactly which parameters are used in training each of the NNs. A table may be helpful in clarifying this. Most importantly, that SA-FNNNO-BIO-2 and Watson et al. (2020) are the only NNs that use salinity and mixed layer depth as predictors.

*Response: We agree with your suggestion and have added a new table (Table 2; Page 7, Lines 152 – 155) which displays the input parameters used in training the respective neural network approaches. Table 2 is referred to in the text at Page 7 Lines 156, 159 and 160, where the SA-FNN variants are described. Table 2 and caption can be seen below:*

***Table 2: The input parameters of the neural network variants described in section 2.3. and 2.6. $xCO_2$ is the atmospheric mixing ratio of $CO_2$.***

| Neural Network Variant | Input parameters |
|---|---|
| SA-FNN$_{NCP}$ | $pCO_{2 \, (atm)}$, SST and NCP |
| SA-FNN$_{NPP}$ | $pCO_{2 \, (atm)}$, SST and NPP |
| SA-FNN$_{CHLA}$ | $pCO_{2 \, (atm)}$, SST and Chl a |
| SA-FNN$_{NO-BIO-1}$ | $pCO_{2 \, (atm)}$ and SST |
| SA-FNN$_{NO-BIO-2}$ | $pCO_{2 \, (atm)}$, SST, salinity, and mixed layer depth |
| W2020 (Watson et al., 2020a) | $xCO_{2 \, (atm)}$, SST, salinity, and mixed layer depth |

Lines 268–270: This sentence is a bit confusing at the moment. One suggested revision here: "This showed that a reduction in pCO2(sw) RMSD of 36% was achieved by eliminating satellite NCP uncertainties, 34% by eliminating satellite NPP uncertainties, and 19% by eliminating satellite Chl a uncertainties."

*Response: We have revised this sentence as suggested; see Page 12, Lines 274-276.*

[Figure]

Registered Office:
Prospect Place
The Hoe, Plymouth
PL1 3DH, UK

T +44 (0)1752 633100
E forinfo@pml.ac.uk
W www.pml.ac.uk
@PlymouthMarine

Patron: James Cameron
Registered charity number 1091222.
PML is a company limited by guarantee,
registered in England & Wales,
company number 4178503

Figure 3: Unfortunately, with the helpful addition of new data to these plots, this figure has become very difficult to interpret (at least given the quality of image I have). This could perhaps be remedied by simply reshaping the panels: an elongated y-axis might help emphasize the distinctions between individual lines. Another option may be to adjust the color palette selection. Or, to split this into two separate figures, showing the climatology from SA-FNNNCP in both.

***Response:*** *We have revised Figure 3 by changing the y-axis limits for each plot as suggested. The figure caption has been updated to highlight the different y-axis limits. The updated Figure 3 and caption can be seen below:*

[Figure]

***Fig. 3:*** *Monthly climatologies of pCO₂ (sw) referenced to the year 2010 for the 8 stations marked in Fig. 1 from the SA-FNN$_{NCP}$, SA-FNN$_{NPP}$, SA-FNN$_{CHLA}$, SA-FNN$_{NO-BIO-1}$, SA-FNN$_{NO-BIO-2}$ and W2020 (Watson et al., 2020b). Light blue lines in Fig. 3a, b indicate the in situ pCO₂ (sw) observations from PIRATA buoys. The atmospheric CO₂ increase was set as 1.5 µatm yr$^{-1}$. Black dashed line indicates the atmospheric pCO₂ (~380 µatm). Error bars indicate the 2 standard deviation of the climatology (~95% interval), where larger error bars indicate a larger interannual variability. Red circles indicate the literature values of pCO₂ (sw) described in section 4.2. Note the different y-axis limits in each plot.*

Line 286: Change "climatology" to "climatologies"

***Response:*** *We have made this suggested change, see Page 14, Line 296.*

Line 363: "…indicated however, that elevated pCO2(sw) at ~430 uatm exist…" During what time of the year is this elevated pCO2(sw) occurring? Year-round?

***Response:*** *The elevated pCO₂ (sw) observed by Bruto et al. (2017) referred to was measured in September for 2008 to 2011, which contrasted with Lefèvre et al. (2020) who reported lower pCO₂ (sw) at ~360 µatm in 2013. We have included the month within the sentence, see Page 17 Lines 372-373, which now reads: "Bruto et al. (2017) indicated however, that elevated pCO₂ (sw) at ~430 µatm was observed in September for 2008 to 2011."*

Registered Office:
Prospect Place
The Hoe, Plymouth
PL1 3DH, UK

T +44 (0)1752 633100
E forinfo@pml.ac.uk
W www.pml.ac.uk
🐦 @PlymouthMarine

Patron: James Cameron
Registered charity number 1091222.
PML is a company limited by guarantee,
registered in England & Wales,
company number 4178503

Line 364: "The PIRATA buoy pCO2(sw) observations (Fig. 3a) clear highlight the difference between these years…" It's not clear to me how the monthly climatology in Fig. 3a highlights a difference between the years. Are PIRATA observations only available from 2008 to 2011, during which time Bruto et al. indicate higher pCO2(sw)?

*Response: The size of the errorbars indicate the interannual variability of the climatology as described in the methods section 2.6., and Figure 3's caption. For September (Fig. 3a), larger errorbars were observed consistent with the differences between Lefèvre et al. (2020) and Bruto et al. (2017). We have updated the sentence to make this clearer, see Page 17 Lines 373 – 375, which now reads: "The errorbars on the PIRATA buoy pCO$_2$ (sw) observations (Fig. 3a) clearly highlight the differences between Lefèvre et al. (2020) and Bruto et al. (2017), but there are less than 4 years of monthly observations available, which do not resolve the full seasonal cycle."*

*In SOCATv2020, data available from the PIRATA buoy at 8 °N 38 °W correspond to the data presented in Lefèvre et al. (2020) and Bruto et al. (2017), which covers 2013 and 2008 to 2011 respectively, but do not provide full annual coverage.*

Line 472: Change "reduced" to "eliminated" or "reduced to zero"

*Response: We have changed this to "reduced to ~0", see Page 21, Line 483, to be consistent with the sentence on Page 16, Line 341.*

Line 475: Recommend revising to "…and two neural networks that do not use…"

*Response: We have revised the sentence as suggested; see Page 21, Lines 484 – 485.*

Line 478–479: Add to the end of this sentence "occurred" or "was observed"

*Response: We have added "occurred" to the end of the sentence as suggested; see Page 21, Lines 488 – 489.*

Registered Office:
Prospect Place
The Hoe, Plymouth
PL1 3DH, UK

T  +44 (0)1752 633100
E  forinfo@pml.ac.uk
W  www.pml.ac.uk
  @PlymouthMarine

Patron: James Cameron
Registered charity number 1091222.
PML is a company limited by guarantee,
registered in England & Wales,
company number 4178503